# Solvent-Producing Clostridia Revisited

**DOI:** 10.3390/microorganisms11092253

**Published:** 2023-09-07

**Authors:** David T. Jones, Frederik Schulz, Simon Roux, Steven D. Brown

**Affiliations:** 1Department of Microbiology and Immunology, University of Otago, Dunedin 9010, New Zealand; 2Lawrence Berkeley National Laboratory, DOE Joint Genome Institute, Berkeley, CA 94720, USA; fschulz@lbl.gov (F.S.); sroux@lbl.gov (S.R.); 3LanzaTech Inc., Skokie, IL 60077, USA; steve.brown@lanzatech.com

**Keywords:** Solvent-producing *Clostridium*, acetone butanol fermentation, biobutanol, prophages, tailocins, phylogeny and phylogenomics

## Abstract

The review provides an overview of the current status of the solvent-producing clostridia. The origin and development of industrial clostridial species, as well as the history of the industrial Acetone Butanol Ethanol fermentation process, is reexamined, and the recent resurgence of interest in the production of biobutanol is reviewed. Over 300 fully sequenced genomes for solvent-producing and closely related clostridial species are currently available in public databases. These include 270 genomes sourced from the David Jones culture collection. These genomes were allocated arbitrary DJ codes, and a conversion table to identify the species and strains has now been provided. The expanded genomic database facilitated new comparative genomic and phylogenetic analysis. A synopsis of the common features, molecular taxonomy, and phylogeny of solvent-producing clostridia and the application of comparative phylogenomics are evaluated. A survey and analysis of resident prophages in solvent-producing clostridia are discussed, and the discovery, occurrence, and role of novel R-type tailocins are reported. Prophage genomes with R-type tailocin-like features were detected in all 12 species investigated. The widespread occurrence of tailocins in Gram-negative species is well documented; this survey has indicated that they may also be widespread in clostridia.

## 1. Introduction

The review includes both a historical perspective and the current status of the solvent-producing clostridia. This encompasses the comparative genomic and phylogenetic analysis of clostridial species and a survey of prophages and R-type tailocins.

In order to provide a general background context to the review, an updated historical account has been included. This covers the origins and development of the industrial solvent-producing species along with the origins, development, and expansion of the industrial Acetone Butanol Ethanol (ABE) fermentation process. The historical information covered in this section has been accumulated by the first author over almost 50 years in the field as both an academic researcher and a consultant and technical adviser to the fermentation industry. The broad historical coverage encompasses an enormous amount of detailed information gleaned from multiple sources. This includes historical information published previously [1] in addition to a large number of other early and recent scientific publications. It also includes material obtained from numerous other sources such as patents, catalogs, culture collections, company sites and reports, books, historical articles and reports, blogs, etc. In addition, a significant amount of information and knowledge has been gained through personal involvement, interactions, communication, and correspondence. This would make the full referencing of these numerous and diverse sources extremely challenging and potentially unwieldy. To make this historic account more readable, references have been kept to a minimum.

The first comprehensive historical account of the ABE fermentation process was published in a 1986 review by Jones and Woods [1]. Many subsequent publications have included brief accounts of the history [2]. At the time that the 1986 review was written, very little information was available in the Western world regarding the large-scale industrial ABE fermentation processes in Russia, Japan, Taiwan, and China. This information is now available in various publications that have been invaluable in providing a comprehensive, up-to-date historical account of the industrial ABE fermentation process [1,3,4,5,6,7,8]. Many of the early publications on the origin of the industrial solvent—producing strains and the industrial fermentation process contain valuable scientific information but very little, if any, historical background. Two publications by Beesch on the industrial acetone-butanol fermentation of starches and sugars give a comprehensive, detailed account of the typical industrial fermentation process [9,10]. The scientific and technical information obtained from numerous articles has been incorporated into the history of the industrial ABE process. However, as these numerous and generally well-known accounts have been extensively referenced, they have not been referenced in this review. A comprehensive historical account of the production of acetone during WW1 was published to mark the centenary of the war [11]. A review by Green gives a useful general overview of recent developments in the field [12]. The second and third sections of the review deal with recent scientific advances and have been referenced accordingly.

## 2. The Origin and Development of the Industrial ABE Fermentation Process

The origin and development of industrial solvent-producing clostridia and the industrial Acetone Butanol Ethanol (ABE) fermentation process are inextricably interlinked. A historical perspective of the industrial fermentation process is fundamental to understanding the historical development of solvent-producing clostridia. The origin of the industrial production of acetone and butanol on an industrial scale began with a quest to produce synthetic rubber. The pressing need to produce acetone for munitions during WWI led to the rapid expansion of the industrial fermentation process.

### 2.1. The Quest for Synthetic Rubber

The shortage of natural rubber that occurred during the early years of the twentieth century resulted in Britain and Germany embarking on a race to become the first to develop and patent a process for the production of synthetic rubber. Two entrepreneurial industrial chemists formed the company of Stranger and Graham Limited (S&G) in London, with factory premises at Rainham Ferry on the Thames Estuary that they used to manufacture varnishes, paints, and cattle feeds. A commercial venture was initiated in 1910 when S&G united with a number of chemists to set up the Research Syndicate Ltd. They contracted Prof. William Perkin, head of an internationally renowned organic chemistry laboratory at Manchester University, to work on the project. Chaim Weizmann, who came to England in 1904, took up a research fellowship with Perkin. Perkin employed Weizmann, and he began working on the development of a process to produce iso-prene as a starting material for synthetic rubber. The main problem was that isoamyl alcohol was not available in the quantities required for industrial production. Weizmann had been a regular visitor to the Pasteur Institute in Paris and developed an interest in bacteriology. He suggested that Prof. Fernbach and his assistant, Moses Schone, be invited to join the team, and in 1910, the Anglo-French Amyl/Butyl Alcohol syndicate was formed. In March 1911, the Fernbach bacillus FB was isolated in Paris, which produced significant levels of butanol from potatoes, and by December 1911, Weizmann had developed a process for rubber synthesis from butanol. After a paper was presented at the Society of the Chemical Industry, Weizmann was highly critical and felt his contribution had been ignored. This led to a fallout with Perkin, and he was dismissed. Weizmann severed all connections with Perkin, S&G and Fernbach and continued to carry out research in Manchester on his own and set about finding a bacterium with the properties he desired. He isolated a culture named the BY bacillus (later renamed *C. acetobutylicum*), capable of transforming the starch of cereals, and in particular maize, into a mixture of acetone and butanol. After natural rubber prices fell, the development of synthetic rubber was no longer economically worthwhile, and he abandoned the project. The Fernbach bacillus was patented in 1911, and another patent was filed in 1912 that covered the formation of acetone from carbohydrate feedstocks fermented using a butylic bacillus of the type Fitz. In July 1912, the Synthetic Products Company Ltd. was launched with considerable opposition from the London rubber interests. Not long after, problems began to emerge for the development of synthetic rubber. The monopoly held by Brazilian natural rubber was undermined by the establishment of Malaysian rubber plantations, and a large amount of capital was invested that greatly increased volume. After Synthetic Products Company was established, the company continued to use the Rainham site until the middle of 1914. In 1912, the company acquired an oil-cake factory located on Alexander Docks in Kings Lynn. This factory was expanded and converted to make acetone and butanol from the starch content of potatoes, and the process was transferred from Rainham to Kings Lynn.

### 2.2. The Development of the Industrial AB Fermentation Process in Britain during WWI

Cordite used in munitions was first developed for military use in Britain in 1889. Acetone was required as the solvent, and at the beginning of WWI, it was produced almost entirely by the destructive distillation of wood. When war broke out, acetone was in short supply, and the loss of Austria as one of the main suppliers added to the shortfall. Britain’s reliance on cordite made a secure supply of acetone critical. In 1914, the Royal Navy suffered a major defeat in the Battle of Coronel off the coast of Chile, largely due to the poor quality of cordite. Winston Churchill, who was the First Lord of the Admiralty, realized that it was essential to make changes and insisted that the Royal Navy should secure its own independent supply of cordite. A decision was made to build a factory for the manufacture of high-quality cordite at Holton Heath, near Poole in Dorset. The site was chosen because it provided both a remote and a secure location. The Admiralty took charge of the construction, and the new factory became the Royal Naval Cordite Factory or RNCF, which began production in 1915. The “Shell Crisis” that occurred in 1915 made it apparent that the war in France could be lost due to the lack of munitions and contributed to the fall of the British Government. A new Ministry of Munitions was established and set about building munitions factories across the country. By 1916, there were 73 munition factories in production, and by the end of the war, the number had increased to 218.

In 1915, Weizmann was asked to demonstrate his process for the transformation of starch into acetone and butanol to the Nobel Explosives Company. He was advised to take out a patent on his process, and Patent No. 4845 was filed in March 1915. In April 1915, Weizmann was summoned to the Admiralty and met with Sir Fredric Nathan and Winston Churchill and visited the Nicholson Three Mills Distillery. As a result of these meetings, Weizmann was given the go-ahead to set up his process. He recruited a group of 15 chemists and bacteriologists and was given space and facilities in the Admiralty Laboratory at the Lister Institute and access to the plant at the Three Mills Distillery. The next step was to establish a pilot plant to test the process on a larger scale, and they succeeded in converting 100 tons of grain to 12 tons of acetone. The success of pilot trials led to the decision to erect a full-scale, purpose-built acetone plant at the RNCF at Holton Heath. By the middle of 1916, full-scale production was started at the Nicholson Distillery.

Meanwhile, in April 1915, the Government entered into a contract with Synthetic Products Co Ltd. to supply acetone produced by fermentation, but difficulties in supply arose right from the start, and the company was never able to produce the minimum quantity of acetone stipulated. As a result, the Government took possession of the Kings Lynn factory under the Defence of the Realm Act in March 1916, and the management was taken over by the Propellants Branch of the Explosives Supply Department. The factory was converted to the Weizmann process with the average weekly production of acetone of just over four tons, which was increased to just over five tons in 1917.

At the beginning of 1916, all alcohol distillers became controlled premises, and it was planned to convert six distilleries to produce acetone. The Nicholson Three Mills Distillery was the first distillery to produce acetone, but it was never that successful or reliable. For the full-scale fermentation process, the existing plant from the distillery was put into use, along with a mixture of adapted and purpose-built equipment. The fermentation tanks were simple iron vats with wooden lids of 136,000 or 178,000 L using a mash that contained 5% (*wt*/*vol*) ground maize as the substrate. The conditions for achieving sterility were far from satisfactory, with seven different steps used to produce the final inoculum, compared with the process at Kings Lynn, with just three steps involved. Despite these shortcomings, Three Mills was still able to produce appreciable quantities of acetone from May 1916 to February 1917. A second acetone plant was also established at the Ardgowan Distillery in Greenock, near Glasgow. The whisky distillery was founded in 1896 and was then converted to making potable grain spirits and industrial alcohol. Of the six distilleries planned, only these two distilleries had become operational when the project was abandoned due to the shortage of imported maize.

The RNCF at Holton Heath was officially opened in January 1916. Larger scale trials at the Three Mills distillery had proved so promising that the Admiralty made the decision to go ahead and erect a full-scale acetone plant at the RNCF at Holton Heath. The acetone plant opened in January 1917 and was the first purpose-built factory to exploit the Weizmann process. The team at Holton Heath that succeeded in getting the process to work consisted of A.C. Thaysen, T.K. Walker, L.D. Galloway and J. Reilly. The acetone factory was the largest building constructed on the site, and the fermentation plant consisted of eight fermenters with a volume of 273,000 L. The maize was milled and cooked and contained around 66% (*wt/wt*) starch. Each fermenter was inoculated with 4.7% (*vol/vol*) inoculum of around 3600 L. Initially, the temperature was not allowed to go above 37 °C, but experiments showed higher temperatures could be tolerated, leading to temperatures of 39 °C to 40 °C being adopted. This shortened the duration of the fermentation process by 2–3 h, but many of the fermentations showed abnormal activity and gave low yields. The fermentation was generally completed within 24–36 h, with some taking up to 45 h. The ratio of butanol to acetone was approximately two to one. The successful operation of the process was dependent on maintaining sterility as far as possible. The plant was designed to produce 2000 tons of acetone per annum (p.a.), but in 1917 produced 3000 tons using both maize and rice. The plant was reliant on maize imported from the US but increasing pressure from the German U-boat campaign disrupted supply, leading to the factory experimenting with other materials, including rice, artichokes, horse chestnuts, and acorns, all with limited success.

By 1918, the need to produce acetone in Britain had declined. The Three Mills Distillery and the Ardgowan Distillery had already stopped producing acetone by the end of 1917, and the factory at Kings Lynn only worked intermittently and was finally closed at the end of 1918. The RNCF continued operating, but production ceased after the armistice was signed. By 1927, RNCF had pioneered a new solventless process for the manufacture of cordite, and the acetone plant was demolished.

### 2.3. The Establishment of the AB Fermentation Process in France during WWI

During the nineteenth century in France, the Cail family founded a beet sugar factory at Melle (Deux-Sèvres, France) called the Cail Sugar Refinery that was later sold and turned into an alcohol distillery, and renamed the Deux-Sèvres Distillery. At the end of 1914, after the distillery was rebuilt following a fire in 1908, it was ordered to increase alcohol production, and later, the factory’s activities were turned over to the manufacture of various organic products needed for the war effort. By 1916, the shortage of acetone for the Allies had become critical. Robert Allenet, the commercial director, traveled to England to investigate the Weizmann acetone fermentation process for potential use in the factory at Melle. Upon his return, tests were undertaken, and cultures were isolated from sewers, drains, and slaughterhouse washings. A spore-forming anaerobe was isolated by Firmin Boinot named *Bacillus butylicus* (BF). The BF bacillus differed from the Weizmann BY bacillus in a number of respects and had the advantage that an inoculum could be transferred from vat to vat indefinitely without any deterioration of function. This resulted in a considerable reduction in time by charging each fermenter with the active wort drawn from the previous fermentation. In June of 1918, Eloi Ricard filed a patent application for the manufacture of acetone that was granted in 1921. The patent and strain were later acquired by the Commercial Solvents Corporation in the US. The factory initially used maize as the raw material, but due to shortages, worts made from rice, wheat, rye, barley, oats, buckwheat, sorghum, cassava, and potato were also used. Worts made from lupin, soybeans, chestnut, acorns, artichoke, and beetroot were only partially successful in producing acetone, as carbohydrate utilization was incomplete. The wort was cooked at 120 °C at a concentration of 20–25% (*wt/vol*) and then diluted with previously heated water to give a final mash concentration of 8–9% (*wt/vol*). The acetone plant consisted of 8 and later 12 cylindrical steel fermenter vessels with a capacity of around 50,000 L. Each bank of four fermenters was served by its own milling system, mash tun, mash boiler, and reservoir. The fermentation was run at 37 °C, and the duration of the fermentation was around 42 h. The ratio of the solvent produced was roughly one part acetone to two parts butanol, and the yield was 23–24% (*wt/wt*) of the weight of the maize. Acetone was one of the key products produced by Ursines de Melle during the war, and it remained at the forefront of acetone production at the national level. The hydrogen gas produced was used to fill tethered balloons and airships. After the war, the company continued to operate a pilot-scale AB fermentation facility, and in 1934, the company was contracted to establish an industrial AB fermentation process for National Maize Products in South Africa, using the Melle cultures. In 1937, Les Usines de Melle became a public limited company, and in 1972, it was bought by the Rhône-Poulenc group and later became Rhodia Food.

### 2.4. The Establishment of the British Acetone Company in Canada

By 1916, the British Government was facing extreme difficulty in obtaining enough acetone for the manufacture of cordite. This resulted in them entering into a contract with Gooderham and Worts and the General Distilling Company in Toronto to utilize part of the plant for the production of acetone by the Weizmann process. As a result, conversion work commenced in May 1916, and the two companies were transformed into British Acetones Toronto Limited. Horace Speakman, a member of the team assembled by Weizmann in London, was sent to Canada by the British Ministry of Munitions along with other technical staff. Other senior staff were seconded from the University of Toronto, and Col. Albert Gooderham was appointed as manager. By the end of 1916, the entire General Distilling Company plant and some parts of the Gooderham and Worts plant were converted at a cost of CAD 1.1 million. The raw material used for the fermentation process was shelled maize. The milled maize mash was sterilized by cooking for 2 h at 100 °C. Initially, a mash concentration of 5% (*wt*/*vol*) was used, but this was increased to a final concentration of 10% (*wt/vol*). Cultures of the Weizmann BY bacillus were initially supplied from Britain. A battery of 16 seed tanks produced the 1000 gallons of inoculum required each day. The full seed culture cycle took seven days to complete. Initially, the fermentation plant consisted of nine 100,000 L fermenters, with another seven fermenters being added in 1917 and a further six in 1918. The fermentable starch content of the maize mash was around 65% (*wt/wt*), giving a solvent yield of around 31% (*wt/vol*), with an acetone yield of around 10% (*wt/vol*). Most fermentations took between 40–46 h, with some runs taking up to 60 h. The solvent ratio was around 60:30:10 butanol, acetone, and ethanol. The products were separated by fractional distillation, and the gases were discharged into the atmosphere. From the beginning, Speakman recognized that aseptic conditions were essential for the successful operation of the fermentation process, and it is remarkable that only 24 fermentations out of a total of 3958 inoculated were lost. By the time the war came to an end in November 1918, the factory had produced around 200 tons per month with a total of 5500 tons of acetone and 7800 tons of butanol. Of this, 2830 tons had been shipped to Britain, amounting to nearly 60% of the total acetone produced. The distillery consumed nearly 4000 tons of coal.

### 2.5. The Establishment of the Acetone Factory at Nasik in India

Professor Gilbert Fowler, a British biochemist at the Indian Institute of Science in Bangalore, was largely responsible for acetone production in India during WWI. While back in England, on leave in 1916, the possibility of establishing an acetone plant was discussed. A decision was made to go ahead and erect a factory, with the cost of building the plant borne by the British Government. Nasik, in the Bombay district, was selected as the site for the factory. Rice had been successfully used in the Weizmann process in England, but a type of sorghum millet known as jowari that contains around 53% (*wt/wt*) starch was found to be a cheaper source of starch. Broken rice and jowari were the main raw materials used, and the grain was milled and cooked at 130 °C. for several hours. The fermentation vessels had a capacity of 90,000 L, and there were two continuous stills. Over 100 tons of acetone were produced before the termination of hostilities. The factory was not commercially viable, and in 1920, it was handed over to the Bombay Government and converted for the production of alcohol.

Australia never utilized the AB fermentation process for acetone production but did develop a microbial process for the production of calcium acetate that was then converted to acetone.

### 2.6. The Establishment of the ABE Fermentation in the United States

When the United States entered the war in April 1917, acetone was still in urgent demand as a solvent for cordite production for the British and was also required for the manufacture of dope for aircraft production by the Americans. After the US entered the war, the British War Mission in Washington investigated the possibility of acquiring a distillery in the US for the purpose of manufacturing acetone. In 1917, the British War Mission purchased the Commercial and Majestic whisky distilleries at Terre Haute in Indiana, situated in the heart of the maize belt, and adapted them for acetone production by the Weizmann process. Information necessary for the construction and operation of the Terre Haute plant was provided by the Canadians, and the bacterial cultures were also obtained from Canada. Production at Plant No. 1 began in May 1918. To manage the enterprise, the Joint War Board set up an incorporated company in New York called Commercial Solvents Corporation of New York. The US Air services purchased a half interest in the company and received half of the acetone and butanol produced. Except for differences in size, the two plants were duplicates of the one in Toronto. The second converted distillery, known as Plant No. 2, went into production in August 1918. Shortly after the Armistice in 1918, operations were terminated, and both plants were shut down, having produced 835 tons of acetone and 1300 tons of butanol. There was no demand for the butanol, and most of it had been stored in a large storage tank as there was no easy way of disposing of it. There was also a large surplus of nitrocellulose in the form of gun cotton. Efforts were made to salvage these products, and representatives from the army approached the chemical industry to try to find a use.

### 2.7. The Establishment of the Commercial Solvents Corporation

After the war ended, the Allied War Board auctioned the plant at Terre Haute in 1919. The plant was bought by a group of businessmen, and in 1920, a new company, Commercial Solvents Corporation (CSC), acquired the assets and secured an exclusive license for the Weizmann process and strains. The plant at Terre Haute became operational in 1920. An adequate market for the fermentation products was needed, but there was only a limited demand for acetone. The butanol that had been stored in a huge steel tank during the wartime operation was found to have potential for use in nitrocellulose lacquers. The US automobile industry had begun a rapid expansion and required a faster method of finishing automobiles. Fast-drying nitrocellulose lacquers were developed, and butyl alcohol and its ester, butyl acetate, were ideal solvents. CSC registered the name Butanol, which rapidly became the accepted name, and demand increased rapidly, transforming the CSC business almost overnight.

In 1922 and 1923, CSC experienced problems with sluggish fermentations that resulted in decreased performance. Fred and Peterson from the University of Wisconsin were among the consultants called in to diagnose and advise on the problems that were caused by bacterial contamination and phage infections. This led to a close working relationship between CSC and the Department of Microbiology at the University of Wisconsin for the next two decades. The ongoing problem with phage infection resulted in a decision to increase the number of fermenters at the Terre Haute plant from 40 to 52. It also led to a decision to decentralize production and to establish an entirely new larger plant. The Majestic distillery at Peoria in Illinois was purchased and converted, and the new plant consisted of thirty-two 189,000 L fermenters, and production began in December 1923. The demand for butanol continued to escalate, and during the period from 1923 to 1926, the capacity of both CSC plants was increased, eventually leading to 148 fermenters in operation at its two plants. In 1927, the output of solvents exceeded 13,000 tons, more than two and a half times the production of two years before, and in 1928, 40,000 tons of solvent were produced. Following on from the work performed on phage infection, an extensive research and development facility was set up at Terre Haute. Although a number of new strains were isolated from various sources, none of these proved to be superior to the original Weizmann strain.

At this time, the raw material used in the AB process was low-grade maize that was otherwise difficult to dispose of. The combined plants utilized between 800 and 850 tons of maize per day. The shelled maize was milled, bran and germ were removed, and the oil was expelled. The maize was mixed with water to give an 8–10% (*wt/vol*) maize mash that was cooked for several hours at 121–126 °C. These were serviced by an equivalent number of pre-fermenters of 1000 gal capacity and culture vessels of 100 gal capacity. The fermentation was run at around 35–36 °C and took around 48 h to complete, giving a solvent ratio of 60:30:10 butanol, acetone, and ethanol. The fermented mash was then run though continuous stills that produced around 50% (*vol/vol*) solvents that were then separated and purified on fractionating columns.

During the 1920s and 1930s, there was a glut of molasses, and the increasing price of maize, coupled with the abundant supply of cheap molasses, provided an attractive alternative to starch-based maize as the raw material. In the early 1920s, CSC had already started investigating molasses as an alternative substrate. The Weizmann patent was due to expire, and CSC was well aware that a number of other US companies were working on the development of molasses-based commercial AB fermentation processes. CSC developed several successful new strains, and the company switched to using cane molasses, except for a period during WWII when shortages required reverting to maize.

Under the protection of the Weizmann patents, CSC was able to maintain its monopoly and dominate the market for butanol. The profits to be made attracted the interest of other companies and provided strong motivation for competitors to establish their own production capacity. CSC maintained an aggressive approach and mounted two legal proceedings against potential competitors. The first litigation was against the Synthetic Products Company in Britain, which had resumed possession of the factory at Kings Lynn and recommenced production in 1920. In 1924, CSC started an action for infringement of the Weizmann patent in the High Court in London, with Elizabeth McCoy retained as an expert witness. The landmark decision, based on the judgement that the FB bacillus and the BY bacillus were different microorganisms, was the first successful patent litigation involving a biochemical process. Not long after, Synthetic Products went into liquidation. The factory site at Kings Lynn was taken over by the Distillers Company, and in 1928, the factory and AB plant were destroyed in an unexplained explosion. The second litigation was against the Union Solvent Corporation in the US. In 1929, the new company began commercial operations at its AB plant near Cincinnati and began selling its products in direct competition. CSC and their licensee entered a lawsuit for infringement, and Elizabeth McCoy was again retained as their expert witness. Before the final outcome of the litigation, a lawsuit for receivership had already been filed against the company by shareholders.

During the 1930s and 1940s, CSC continued to expand and diversify its business activities into other areas of industrial chemicals, including pigments, carbon blacks, nitro-paraffins, plasticizers, and explosives, and it also produced ethanol and potable alcohol. In addition to the primary fermentation products, they also made various derivatives and pioneered the process of converting hydrogen and carbon dioxide into synthetic methanol and ammonia and began operations in high-pressure synthesis chemistry using natural gas. CSC also expanded into the production of riboflavin, animal feeds, animal health and nutrition products, agricultural chemicals as well as antibiotics. In 1975, CSC merged with the International Minerals and Chemical Corporation and subsequently bought Pitman-Moore Inc. In 1987, the Terre Haute plant changed its name to Pitman-Moore, and in 1998, Schering Plough bought out Pitman-Moore, and the name was changed to Schering Plough Animal Health. The various Terre Haute plants were permanently shut down in January 2000.

### 2.8. The Development of Industrial Clostridial Strains for Fermenting Molasses

The availability of molasses as a less-expensive and more convenient sugar-based raw material had many advantages. It was potentially cheaper to process, it provided a more fluid sugar-based medium that permitted the fermentation of a higher concentration of carbohydrates, and it halved the cost of distillation. The molasses produced as a by-product of the cane sugar industry was readily available as blackstrap molasses, as well as high test molasses, condensed directly from cane juice.

At CSC, attempts were made to substitute part of the maize mash with molasses, along with the addition of soybean meal and other protein sources, to provide the additional nutrients required. These trials resulted in very extended fermentation times and reduced yields, and little success was achieved. Although extensive trials were continued using existing starch-fermenting strains, it became apparent that an economically viable molasses fermentation process was never going to be achieved using the existing industrial strains of *C. acetobutylicum*. The company had an excellent research department, and the search for new, more effective isolates of solvent-producing clostridia that would utilize molasses was initiated.

In 1926, Elizabeth McCoy isolated 26 solvent-producing clostridial strains that were made available to CSC [13]. The existing culture collections were screened, and a number of strains were identified that were able to ferment molasses sugars to varying degrees. One of the new cultures was able to carry out the fermentation in media containing 6% (*wt/vol*) sugar and became the famous CSC No. 8 strain isolated by McCoy. This strain was patented by both CSC and McCoy as *C. saccharo-acetobutylicum* (now *C. beijerinckii*) and became the first of many improved strains of this type that were capable of fermenting molasses without the requirement for the inversion of the sucrose, at sugar concentrations of around 6% (*wt/vol*), giving solvent yields approaching 30% (*wt/vol*). By 1932, CSC had established a commercially viable AB fermentation process using molasses. The No. 8 industrial strain appears to have still been in use around 1935 as the main production strain for the fermentation of molasses by both CSC in the USA and at their newly opened fermentation plant in the UK. This strain was superseded a few years later by a new generation of superior industrial saccharolytic clostridial strains. During the period 1936 to 1940, CSC filed several US patents covering the industrial production of acetone and butanol from molasses. The next successful industrial saccharolytic strains were patented by CSC under the name of *C. saccharo-butyl-acetonicum-liquefaciens* (now *C. saccharobutylicum*) that consisted of two variants (gamma and delta) described in patents filed in 1936, 1937, and 1938. These new strains allowed shorter fermentation times and produced solvent yields of 30–33% (*wt/vol*), with improved solvent ratios and enabled sugar concentrations of 6.5% (*wt/vol*) and above to be routinely used. Following their introduction, these strains were utilized as the main production strains by CSC. In 1940, CSC patented another industrial strain under the name of *C. granulobacter acetobutylicum* (now *C. beijerinckii*). The performance of this strain was similar to the *C. saccharo-butyl-acetonicum-liquefaciens* strains. Laboratory trials undertaken using optimized media reported solvent concentrations of up to 25 g/L, with yields of up to 32% (*wt/vol*), obtained after 60–70 h.

In addition to strains that produce acetone and butanol, all four American companies that established the ABE fermentation process also either patented or used strains that were probably *C. beijerinckii* that produced mainly butanol and iso-propanol. Similar strains were also isolated and developed in Japan and were used in Taiwan. Once companies ceased operating the fermentation process, most of these industrial strains were lost. Only a few of the surviving strains have been maintained in various culture collections. The CSC industrial strains could well have suffered the same fate. However, examples of these strains were provided to National Chemical Products (NCP) in South Africa in 1944–1945 as part of the allied war effort, and when NCP ceased operations, examples of these production strains were saved.

## 3. The Worldwide Expansion of the Industrial AB Fermentation Process

From the early 1930s, the industrial AB fermentation process underwent a rapid worldwide expansion and was second only to ethanol fermentation in importance. During WWII, it once again played a key strategic role, and further expansion continued after the war, but the commercial process went into decline due to the competition with cheaper petrochemical-based solvent production. In the state-run communist economies in the Soviet Block and China, the fermentation process continued to operate for the strategic production of solvents.

### 3.1. The Expansion of the Industrial Fermentation Process in the United States of America

After the CSC patent lapsed, two major US fermentation and chemical companies entered the market and built plants to produce solvents by fermentation. Publicker Industries, located in Philadelphia, and US Industrial Chemicals Inc., located in Baltimore and New Orleans, were already major producers of ethanol. Along with CSC, these three companies became major contributors to the rapid expansion and dominance of the chemical industry in the US. From the early 1930s, all the US companies operating the ABE fermentation process switched to the use of sugarcane molasses as the preferred fermentation substrate due to its ready availability and lower cost. All four of the US companies that operated the ABE fermentation process also used the process to produce riboflavin and marketed a wide range of animal feed products as a valuable by-product of the fermentation. During WWII, these companies were again major contributors to the war effort. During the war, the fermentation industry in the US was required to switch back to the use of maize as the substrate due to limitations in accessing and transporting molasses. After WWII, the major advances made in the petrochemical industry facilitated the production of solvents by petrochemical processes. This, coupled with the low cost of the oil-based raw materials, resulted in strong competition being exerted on the US fermentation industry, and the use of the ABE fermentation process was gradually phased out in the US.

Publicker Industries Inc. began operating a distillery on the Philadelphia riverfront in 1912 and was incorporated in 1913. It became a public company in 1946 and was one of the most successful alcohol producers and distillers in the US in the mid-twentieth century, and it continued to produce whiskey, potable spirits, industrial alcohol, and other chemical products until 1985. The company began operating a large-scale ABE fermentation process from around 1932 until 1977. Initially, the process used maize and converted to molasses in 1936. Their butanol process utilized at least three main industrial strains. These were patented under the names of *C. saccharo-acetoperbutylicum* (US patent 2,420,998) and *C. amylo-saccharobutyl-propylicum.* (US patent 2,439,791). The Publicker process used spherical fermentation tanks and had five Horton spheres of 7.57 million L, ten of 1893 million L, plus sixteen of 68,000 L. By the mid-1950s, the main industrial chemical plant located on the Delaware River in Philadelphia had developed into one of the largest distilleries in the world and is said to have had the world’s largest capacity for ethanol production and extensive facilities for converting stillage to animal feeds. From the 1950s, Publicker Industries started on a decline that continued for the next four decades, and it finally ceased operations in 1985. During demolition, the site was discovered to contain large residues of toxic and hazardous wastes that involved Publicker in protracted environmental clean-up litigation.

The US Industrial Alcohol Company (USI) was incorporated in West Virginia in 1906. At this time, the company had four ethanol distilleries that utilized molasses as the raw material. To meet the increased demand for ethanol, USI acquired and built a new plant at Curtis Bay in Baltimore that began operations in 1916. The new plant had a battery of 1000 generators to produce acetic acid that was then converted to calcium acetate to make acetone for munitions. In the decade following WWI, the company continued to expand, purchasing additional plants in Peoria, Illinois, Newark Bay, New Jersey, Anaheim, California, and Westwago, Louisiana, and it became one of the global leaders in the industrial chemical industry. At the beginning of the 1930s, the company embarked on the production of butanol. Two ABE plants were built; the first at Curtis Bay began production in 1931, and the second at the New Orleans site began production in 1935. The plant at New Orleans was successful in developing an ABE process for producing butanol from molasses, and the process was then installed at Curtis Bay. The company developed and patented two industrial saccharolytic *Clostridium* strains for butanol production, *C. vicifaciens* in 1935 (US 2,017,572) and *C. celerifactor* in 1939 (US 2,169,246). A large range of stock and poultry feed products made from the stillage from the ethanol and butanol fermentation processes were developed by the company. In 1943, the company was merged into US Industrial Chemicals Inc. At the end of the war, the company converted surplus potatoes and cassava, imported to assist Brazil, to produce ethanol and butanol. In 1948, a large-scale fermentation pilot plant was built at Curtis Bay that was used for optimization studies on ABE fermentation. The company merged with National Distillers Products in 1951 and changed its name to National Distillers and Chemical Corporation. Around this time, the company moved away from producing ethanol and butanol from molasses. In 1954 the company sold the Curtis Bay plant and in 1957, the company became a division of the Quantum Chemical Corporation.

The Western Condensing Company was established in San Francisco in 1918 by David Peebles, a noted food engineer, inventor, and entrepreneur. The company specialized in the production of milk and whey products, including specialty feed products for animals. The company established an Eastern Division Office at Appleton, Wisconsin, in 1939 that included an ABE fermentation plant. The company filed several patents relating to the synthesis of riboflavin using solvent-producing clostridia, one of which referred specifically to the use of *C. acetobutylicum* for this purpose. Other patents issued cover the separation of iso-propanol from butanol and indicate that they were also using a saccharolytic *Clostridium* strain in their processes that was probably a strain of *C. beijerinckii.* In 1952, the company was operating 28 plants engaged in the processing of milk and the manufacturing of milk by-products and was the largest producer of whey products and milk sugars in the world. In 1956, the Western Condensing Company merged with Foremost Foods. The plant in Appleton continued to produce various animal feed products until it closed around 1970.

The Eastern Alcohol Corporation, located in Philadelphia, was jointly owned by National Distillers Products and Dupont de Nemours and Company. In 1929, US patents were taken out on solvent-producing *Clostridium* strains for use in the ABE fermentation process (US 1,725,083) and another in 1933 (US 1,908,361). There is no evidence that the company operated a commercial-scale ABE fermentation process.

The A. O. Smith Corporation was an engineering company based in Milwaukee. In 1933, the company introduced steel beer barrels with a special liner that protected the beer from metallic migration and helped revitalize the Milwaukee brewing industry. Using its innovative glass-fused-to-steel technology, the company developed large-volume, one-piece, glass-lined fermentation tanks to meet the demands of the brewing industry and storage tanks for other beverages and chemicals. Between 1935 and 1941, James F. Loughlin lodged eight US patents assigned to the A. O. Smith Corporation involving the ABEl fermentation process. These included the patenting of three new saccharolytic solvent-producing clostridial strains to produce acetone and butanol from molasses, lodged under the names of *C. saccharobutyl-acetonicum* (US 1,992,921), *C. saccharobutyl-iso-propyl-acetonicum* (US 2,096,377) and *C. saccharobutyl-iso-propyl-acetonicum-beta* (US 2,219,426). Although the company had links with the Schultz Brewery, there is no evidence indicating that either of these companies was involved in operating a commercial ABE fermentation process.

### 3.2. The Establishment of the Industrial Fermentationprocees in Puerto Rico

In the 1920s, the unincorporated US territory of Puerto Rico, in the northeast Caribbean Sea, was a densely populated island with little industrial development, and it depended almost exclusively on its agriculture for the subsistence of its inhabitants, aided by Government support. By the 1930s, worldwide economic and political developments acted together to bring agriculture on the island to the brink of a crisis. The Puerto Rico Reconstruction Administration, which operated from 1935 to 1955, was an agency established as part of the New Deal implemented by the administration of President Franklin D. Roosevelt. The Agency established a sugar program that included the purchase of two sugar mill estates and provided a loan of USD 550,000 for the construction of an AB fermentation plant adjoining the sugar mill. This elicited a number of sharply critical responses in the US, questioning the use of American taxpayers’ money to create unfair and uneconomic competition for American industry. The plant was built by the Lummus Company of New York, an organization that specialized in industrial engineering. Lummus filed a patent application (US 3,715,3640A) on their process technology for the new plant in 1940. Their patent covered the use of blackstrap molasses for the fermentation by *C. madisonii*, which was licensed by the Wisconsin Alumni Research Foundation. Initially, the fermentation used *Bacillus teryl* on invert sugar, isolated by Rafael Arroyo, the chief research chemist at the Puerto Rico Experiment Station. Elizabeth McCoy filed a patent application for *C. madisonii* in 1940 that was granted in 1946. During the early 1940s, the AB plant suffered from serous bacteriophage infections, and the department at Wisconsin was involved in attempts to find solutions for the problem. A report was published in 1943 describing four different types of phages isolated from various infections. The plant had two fermenters of approximately 280,000 L and used Puerto Rican black strap molasses with fermentable sugars set at 5–6% (*wt/vol*) and was also able to use high test molasses. The fermentation was run at 30–32 °C, and the solvents were recovered and purified by conventional distillation. The plant came into production in 1939, and at full capacity, it produced more than 2500 tons of solvents p.a. By 1944, the plant had produced substantial profits and continued operating into the 1950s.

### 3.3. The Transfer of the Fermentation Process to the UK

In England, the Distillers Co. made a reassessment of the process on a pilot plant scale in the late 1920s but concluded that the process was not economical under the circumstances prevailing at the time. In 1934, a joint venture between CSC and Corn Products was established to form a British subsidiary, Commercial Solvents—Great Britain. A new plant was built at Bromborough across the river Mersey from Liverpool that had harbor facilities for the largest tankers of the time at the adjacent docks at Port Sunlight. The cost of importing molasses by sea was less than the long overland haul to the two CSC U.S. plants in Indiana and Illinois.

The factory was designed to produce both industrial alcohol and acetone and butanol, using separate fermentation plants and distillation units, and was manned by British-trained personnel under the leadership of J. Hastings. After minor teething troubles, the solvent process became operational at the end of 1935. The plant initially used the CSC number eight strain and later switched to a phage-immunized strain of C. *saccharoacetobutylicum* developed at Terre Haute. Hydrogen from the AB fermentation was used for the hydrogenation of edible oils. In 1938, the CSC management decided the future of their business interests lay in America, and the plant was sold to the United Molasses Company. Shortly after, it was resold to Distillers Company Limited in the UK (DCL), and the name was changed to Commercial Solvent Ltd.

The process used imported molasses diluted to around 5–7%(*wt/vol*) fermentable sugars (with 6% (*wt/vol*) most commonly used). It was sterilized by heating to 125 °C for 2 h. The seed culture build-up consisted of three stages. The batch fermentation carried out in a bank of 250,000 L fermenters took between 36 and 48 h. The ratios of solvent obtained varied between 65–68% (*wt/vol*) butanol, 30–33% (*wt/vol*) acetone, and 2–5%(*wt/vol*) ethanol. Solvents were originally separated and purified by batch distillation, but later, continuous distillation became the method of choice. Hydrogen from the AB fermentation was used for the hydrogenation of low-melting edible oils for margarine manufacture. Carbon dioxide from both the AB fermentation and the ethanol fermentation was recovered and sold as dry ice or compressed gas. When WWII started, ethanol production from molasses virtually ceased in the UK, but due to the high demand for acetone, ABE fermentation was given top priority. As the war progressed, the demand for acetone again rose to a very high level, and the fermentation plant at Bromborough was expanded. The fermentation capacity was further increased by sequestering suitable alcohol fermenters. This resulted in production levels being achieved that were far beyond the capacity of the original batch-distillation plant. In addition, successful semi-continuous methods for fermentation were devised, which gave slightly lower yields but cut the fermentation cycle to 30–32 h. After the war, the plant at Bromborough endeavored to restore normal commercial operations, and the plant was enlarged again in 1946. In the 1940s, DCL took a substantial shareholding in National Chemical Products (NCP) in South Africa. As a result, DCL made available their technical experience and know-how relating to the production of organic chemicals. Joint comparative studies were undertaken both at the DCL Research Department at Epsom and at the NCP Research and Development Department. By the 1950s, the Bromborough plant faced an increasing challenge from chemically produced solvents, and due to the increasingly high price for molasses, the plant switched to using home-produced beet molasses. Finally, it was decided that the fermentation process was no longer competitive, and the plant ceased operations in 1957. In 1959, DCL wound up Commercial Solvents, and in 1965, DCL transferred its NCP shares to a wholly owned subsidiary, Distillers Chemical and Plastics Limited. This company was then sold and subsequently became BP Chemicals (UK). In 1967, Sentrachem, the South African holding company of NCP, acquired all BP Chemicals shares. DCL continued to occupy the commercial solvent works site until 1977, after which the site was sold and converted to a dye manufacturing works. The buildings associated with these works were later demolished.

### 3.4. The Industrial Fermentation in South Africa

In 1935, a commercial ABE fermentation process was established in South Africa by National Maize Products, later renamed National Chemical Products, utilizing French technology and strains. The factory was situated in Germiston in the heart of the maize-growing area and also produced ethanol and acetic acid as well as solvents. The ABE fermentation process employed the standard batch process and consisted of a bank of twelve 100,000 L fermenters using maize as a substrate. During WWII, the company was contracted to produce acetone for the manufacture of cordite for use in munitions. The rising price of maize, coupled with its importance as a strategic raw material, resulted in the company being ordered to switch to molasses as the fermentation substrate. During 1944 and 1945, molasses fermenting strains were supplied to NCP by CSC in the US and later by CS-GB in England. In 1944, NCP became a public company, and DCL took a major shareholding. In 1967, the South African holding company Sentrachem acquired all shares in NCP, and control reverted to South African ownership. In addition to the production of solvent, the gases were recovered, the water was recycled, and the stillage was combined with ethanol stillage to produce a profitable line of animal feeds. The molasses fermentation ran smoothly from 1946 until shortages of molasses coupled with increasing competition resulted in the ABE fermentation process ceasing operation in 1983. In 1997, the Dow Chemical Company acquired control of Sentrachem and its subsidiary, which were sold off, and the Germiston plant was demolished.

### 3.5. The Strategic Development and Production of Butanol in Japan

With a long history of fermentation practices, Japan began investigating the commercial possibility of the production of acetone and butanol at the beginning of the 1920s. Work was undertaken by the Agricultural Department of Tokyo Imperial University. Bacterial cultures, including the Weizmann BY strain, were imported from the UK, and cultures of solvent-producing clostridia were also isolated and selected locally. Work was also undertaken in Manchuria by the Manchurian Railway Company that included the isolation of the Rokushu No. 1 strain dating from 1922 and the BGP strains dating from 1924 and 1930. Hokkaido, the northern-most island of Japan, had nearly one-fourth of Japan’s total arable land and ranked first in the nation in the production of a range of agricultural products that included wheat, soybeans, potatoes, and sugar beet. In 1927, investigations into the use of the industrial ABE fermentation process were initiated at two research institutes in Hokkaido, where sweet potatoes were identified as a potential fermentation substrate. In 1928, studies on industrial ABE fermentation were also begun in the Agricultural Department of Kyushu University. A number of culture collection strains were tested, and the most promising strains were selected for pilot-scale trials, including *B. butyloacetonicus*, patented in 1933. Although the trials had produced favorable results, the process was not commercialized at this time. In 1931, an industrial-scale ABE fermentation plant was established in the Osaka area by the Kyoei Company in cooperation with the Saki-shi Kondo Pharmaceutical Company. The initial production of butanol was 20 tons per month, but by 1933, production had doubled to 40 tons a month. In 1935, Japan produced 1000 tons of butanol, and this increased to 2000 tons in 1937.

With only limited natural resources, Japan was reliant on indispensable imports. This vulnerability became a major driving force for Japan’s militaristic government, embarking on a mission of territorial expansion, which brought about WWII in the Pacific. The Japanese were well aware of their dependence on oil and, in particular, their reliance on the US for the importation of crude oil, petroleum, and aviation fuel. With the aim of achieving independence in liquid fuel supply, Japan embarked on a dual strategy of gaining access to the oil fields in Southeast Asia and establishing a synthetic fuel industry. Research into the strategic production of synthetic fuel was initiated in the 1920s. The Japanese Army and Navy had different imperatives, and each service developed its own air force. Both services realized that aerial warfare would not only depend on the ability to design and build aircraft and train pilots but would be dependent on the capacity to produce high-performance aviation fuel. To the detriment of the Japanese war effort, neither service cooperated with the other, and each independently developed its own aircraft and sources of aviation fuel. The Japanese Navy initiated research into synthetic aviation fuel production, including the use of ethanol produced by fermentation; however, several drawbacks were recognized in using ethanol as aviation fuel. The Navy also initiated a program to investigate the production of butanol by fermentation as another potential candidate for aviation fuel.

In 1936, three Japanese alcohol distillers, Takara Shuzo, Godo Shusei, and Dainippon Shurui, amalgamated to form a consortium that specialized in the production of ethanol and other fermented products. In 1937, the Kyowa Hakko Kogyo company founded the Kyowa Chemical Research Laboratory in Tokyo. The same year, the Japanese government passed the Alcohol Monopoly Law, which empowered it to take appropriate action to ensure the maintenance of a stable supply of liquid fuels. The government selected the Kyowa Chemical Research Laboratory in Tokyo to develop the technology required to enhance octane levels in aviation fuel. The government contract specified developing the chemical technology for the synthesis of iso-octane by alkylation and the polymerization of butanol. This required the production of large volumes of butanol and led to the rapid expansion of the industrial ABE fermentation process. Independently, around 1942, the Japanese army also initiated research at the Army Fuel Research Centre in Tokyo on processes to produce iso-octane. The Godo Shusei Company, based primarily in Hokkaido, was selected as a location for the production of butanol. The ABE fermentation technology developed by the Hokkaido Institute of Industrial Technology was transferred to Kyowa Chemical Research Laboratory and, in conjunction with Godo Shusei, an ABE fermentation plant was built at their factory site in Asahikawa in 1938. This plant used both potatoes and maize as the major fermentation substrates. In 1939, another ABE fermentation plant associated with these companies was established in Hachinoe in the Aomori Prefecture, located on the northern tip of Honshu. The Hachinoe alcohol plant consisted of twelve 90,000 L fermenters and was converted to produce acetone and butanol in 1939. This plant used potatoes and sweet potatoes as the fermentation substrate and produced around 1500 tons of solvents per year. Godo Shusei took over the former Tohoku alcohol factory and secured a contract to provide the Japanese Army with butanol and acetone. Another industrial ABE plant was built around this time, situated in the vicinity of Nagoya, constructed under the direction of the Kyowa company that consisted of ten 90,000 L fermenters. A further 10 fermenters were added later, and the solvents were recovered by continuous distillation that produced around 8% of Japan’s total production of butanol later in the war. It appears that there was a second butanol plant located at Yokkaichi near Nagoya, operated by the Dainippon Shurui Company, that began production in 1942. This is probably the plant that supplied butanol to the Japanese Navy’s second fuel depot in Yokkaichi. Takara Shuzo, the third alcohol production company in the Kyowa consortium, was based in Kyoto and might have been associated with the initial ABE plant located near Osaka.

The Japanese Navy established its own supply facilities, consisting of six naval fuel depots associated with designated naval districts. The use of butanol for iso-octane production led to the building of a commercial-scale plant erected for the Navy at the second naval fuel depot at Yokkaichi. This industrial facility, named the Japan Butanol Company, was operated by Kyowa Sangyo and began producing butanol in 1942. In 1944, US intelligence estimated that Japan was aiming to produce 75,000 tons of butanol, which would have been sufficient to produce three million barrels of high-octane aviation fuel. Butanol was used both as a source for iso-octane production and was also added directly into aviation gasoline. The production of butanol and acetone more or less doubled every year during the four-year period from 1940 to 1943 but declined sharply by the end of 1944 due to the importation of raw materials being halted. Widespread problems with phage infections were reported at a number of factories that affected both the sugar-based and starch-based fermentations. A 1945 US intelligence report states that Japan had 19 butanol plants of unknown capacity and uncertain location, and it provided estimates of capacity for five plants known to be located in Taiwan as 3.2 million gal p.a.

Japan also built an ABE fermentation plant at Yanji in Manchuria, which mainly used maize as the substrate. Japan also attempted to establish ABE plants in Java and the Philippines under the direction of the Godo Shusei company, which were planned to utilize sugar cane. From the limited information available various difficulties and teething problems were encountered, and operations were short-lived, so little or no butanol was produced.

### 3.6. The Postwar Industrial Process in Japan

After the war, the US occupation authorities forced many Japanese businesses to undergo restructuring. Amongst these was the Kyowa Hakko company that resumed commercial activities and, in 1948, used molasses or sugar syrup as the substrate. The factory was located at the company’s plant in Yokkaichi and operated for about 14 years, from 1948 until around 1961, when the fermentation process was superseded by a synthetic process for the production of solvents. There is little information regarding the details of this plant or its operation, but phage infections remained a problem. At least one other Japanese company operated a commercial ABE fermentation process. The Sanraku Distiller’s Company established a plant at their Yatsushiro plant in the Kumamoto prefecture sometime during the 1950s. Again, there is very little information regarding the details of this plant or its operation. Hongo and Nagata isolated and patented a new high-butanol-producing *Clostridium* from the soil in 1959 and named it *C. saccharoperbutylacetonicum*. The new isolate proved to be very efficient but was found to be susceptible to phage infection. Phage contamination occurred twelve times during one year, and these setbacks appear to be partly responsible for the process being abandoned in the early 1960s. Hongo, Ogata, and their colleagues continued to study these phages and associated Clostocins for most of the next two decades and published numerous papers in this area. A 1963 worldwide survey of fermentation industries prepared for the International Union of Pure and Applied Chemistry reported that in 1963, Japan’s annual production of acetone-butanol by fermentation was around 15,000 tons.

### 3.7. Butanol Production in the Japanese Colony of Taiwan

Taiwan was ceded to Japan in perpetuity in 1895 as part of the post-war settlement following the First Sino–Japanese War. Japan embarked on the rapid modernization and industrialization of the island with a well-developed infrastructure. The island produced a large range of agricultural products and had significant coal deposits along with some oil and natural gas deposits. An extensive sugar industry was developed, which produced around 17 million tons of sugar annually, and in 1939, Taiwan was the seventh largest sugar producer in the world and had become a major exporter. There were eleven sugar companies, and alcohol plants were located adjacent to most sugar factories and refineries. Both sugar and starch-based agricultural crops were used for ethanol production, and it also became a well-established export industry. Its natural resources made the island an obvious choice for the expansion of the Japanese ABE fermentation industry. In addition, a central research Institute was established in 1921, and Taipei Imperial University had well-established linkages with Japanese universities. Taiwan had also been developed as a major Japanese military base, and the navy operated extensively out of the ports around the island.

The city of Kagi (Chiayi) in southern Taiwan was selected as the site of the initial butanol plant because it was a regional center for the production of sweet potatoes. The plant was constructed in 1939 specifically to manufacture solvents by fermentation. The plant employed the classic ABE batch fermentation process, utilizing mainly sweet potatoes, along with some maize and sorghum, using derivatives of the original Weizmann strain and other similar Japanese isolates. The raw material was brought in from the surrounding countryside by a narrow-gauge rail network. At full capacity, the plant consisted of 92 fermenters, each with a capacity of 170,000 L, plus 6 fermenters with a capacity of 85,000 L. The fermenters were inoculated from eight mid-sized pre-fermenters, and the solvent was recovered and separated by distillation. The factory was reported to have employed 3200 people and was the biggest of its type in the Far East. One-fifth of the population of Kagi depended on it for employment, and much of the surrounding country was devoted to growing sweet potatoes for the factory. Initially, the factory was owned and operated by the Taiwan Development Corporation. In 1942, a joint arrangement was entered into with the Japanese Navy, and the name of the plant was changed to the Taiwan Development Chemical Industry Company. At this time, production was 1500 tons. Japanese naval personnel undertook research on the optimization of the butanol fermentation process, and in 1943 and 1944, production was increased to around 5500 tons a year, although the main fermentation continued to utilize sweet potatoes. In 1943, the plant was adapted to using sugar, and in 1944, it switched to using cane juice and syrup. The plant did not have a sugar milling facility, so raw materials were brought in by rail. During the last months of the war, the plant was converted to producing ethanol.

With the advent of the war, increased production of butanol was given a high priority. Although large volumes of sugarcane-based raw material were available, this could not be used due to the lack of suitable saccharolytic strains. In 1941, a program to develop a butanol fermentation process using sugar cane as the raw material was given top priority. The Southern Research Institute coordinated the program involving various institutions in Taiwan, including the Industrial Research Institute, the Sugar Research Institute, the Sugar Research Society, the Taipei Imperial University, and the Taiwan Monopoly Board. Around 40 new strains were tested, and success was finally achieved by Professor Baba at the Taipei Imperial University with an isolate named *C. toanum*. A large-scale pilot plant at the University was used in this development. Another strain isolated in Japan around this time was also patented and described in the Japanese scientific literature in 1943 as *C. butanologeum*. The process used a set sugar concentration of around 7% (*wt/vol*) supplemented with 7% rice bran. Around 100 tons of sugar substrate produced around 16 tons of butanol, 12 tons of iso-propanol, 1.5 tons of acetone, and 0.5 tons of ethanol with a yield of around 30% (*wt/vol*). Fermentation times for the new sugar-based process were reduced to almost half, and the batch fermentations were completed in 30–40 h.

Under the direction of the Taiwan Monopoly Bureau, a program for the mass production of butanol from sugar cane was implemented. The four main sugar companies, Taiwan Sugar, Meiji Sugar, Nitto Development, and Ensuiko, established a cooperative unit called Taiwan Synthetic Fuels. A number of new butanol plants were constructed during the years 1942–1944 associated with existing sugar mills, sugar refineries, and ethanol plants. More than half of the country’s 30 alcohol production plants were also converted to produce butanol. Various wartime US intelligence reports identify at least ten butanol plants situated at Heito, Hinchu, Kagi, Kieshu, Kobi, Kyoshito, Mato, Shoka, Takoa, and Taichu. The plant operated by the Taiwan Development Company, located close to the oil refinery and deep-water port in Takao, had a capacity of about two-thirds that of the Kagi plant. The Sixth Naval Depot’s new butanol fermentation plant at Hinchu (Shinchiku, Japan), designed for the production of iso-octane, was completed in 1943. The plant was reported to have a reactor capacity of 10 million L and began producing butanol in 1944. The second phase of the project to produce iso-octane had still not been completed by the time the plant was destroyed by USAAF bombing. The USAAF bombing campaign began in January 1945 and ended in July. During this period, 30 sugar mills, refineries, and alcohol/butanol plants were bombed. The bombing campaign was credited with the destruction of at least 75% of the plants. The report from the Japanese Governor General Office listed 17 plants as completely destroyed, 9 moderately damaged, and 4 slightly damaged.

### 3.8. The Postwar Industrial Process on the Island of Taiwan

After the war ended, the Republic of China became responsible for governing the Island of Taiwan. During the next two years, all physical assets on the island to do with industry, trade, commerce, and transportation were nationalized. The four largest private sugar companies were amalgamated into the Taiwan Sugar Company, and the six oil companies were merged into the China Petroleum Corporation. Kagi was renamed Chiayi, and the rebuilding of the butanol plant on the original factory site was started in 1946. The Chiayi Solvent Works was established in 1947 and went into full production using sweet potatoes and cassava as the substrate, producing solvent in the ratio of around 56% (*wt/vol*) butanol, 30% (*wt/vol*) acetone, and 14% (*wt/vol*) ethanol. The initial export of butanol to Britain began in 1948. In 1949, the Chinese Petroleum Corporation moved its headquarters to Taipei, and the Chiayi Solvent Works was placed under its control. The increasing price of raw materials for the fermentation process resulted in a gradual switch from sweet potato to cassava and then to molasses. The plant used two main strains, designated the C and the T strains, for the fermentation of cane molasses. During the period from 1952 to 1954, Baba published a series of seven papers relating to the use of *C. toanum* for the optimization of the butanol iso-propanol fermentation process using blackstrap molasses, sugar cane juice, sugar syrup, and raw sugar as the raw materials. Blackstrap molasses became the major fermentation substrate, with fermentable sugar concentrations varying between 7.0 and 7.3% (*wt/vol*), giving solvent yields of 30–33%. The standard additives to the mash included 0.3% rice bran, and the batch process was maintained at a temperature between 33–37 °C and completed after 30–40 h. The solvent ratios were strongly influenced by the nature of the fermentation substrate and varied between 53–65% (*wt/vol*). butanol, 19–44% iso-propanol, and 1–24% acetone. Solvents were separated and purified by continuous distillation. The process was finally abandoned after having operated for a period of 19 years.

### 3.9. The Establishment and Development of Industrial Fermentation in the USSR

The industrial ABE fermentation process is reported to have been first established in the USSR during the period 1929–1935. The industrial process to produce acetone and butanol was developed independently from Western countries and was based on the technology developed by VN Shaposhnikov and colleagues. *Clostridium* strains, classified as *C. acetobutylicum*, were isolated by FM Chistyakov and were reported to have the useful features of being resistant to bacteriophage infections and were capable of high productivity at around 37 °C. The initial process developed was batch fermentation, and the common starch-based fermentation substrates included wheat, barley, rye flour, and potato starch. Some of the plants were located in major sugar beet growing areas, and beet molasses was used as the feedstock. There were eight industrial AB fermentation plants built and operated in the USSR. Four of these plants were located in the Central Economic Region to the south of Moscow. These included the Efremovski plant at Efremov in the Tula Oblast; the Bulchovski plant at Bolchov in the Oryol Oblast; the plant at Mitrofanovka in the Voronezh Oblast; and the plant at Michurinski in the Tambov Oblast. There was another plant located in Petrovski in the Saratov Oblast in the Volga Economic Region. One other plant was located further to the east in the Ural Economic Region at Talitsa, in the Sverdlovsk Oblast. The remaining two plants were in the Caucasus in the far south. The Dokshukinski plant was located in Nalchik, founded in 1913 at the railway station at Dokshukino and later renamed Nartkala. Nalchik is now within the Karbardino-Balker Republic. The second plant was at Grozny, the capital city of Chechnya in the Chechen Republic. The plant in Michurinsk was said to be the first plant to produce acetone and butanol in Russia, and the AB plants at Grozny and Dokshukinski were both in production before WWII. No information has been found regarding when the other plants were built and for how long they remained in operation, but it is probable that some of these plants were only constructed after WWII. At least some of the plants continued to operate up until the late 1980s, but all were closed by the time of the dissolution of the USSR in 1991. The biggest among the Russian plants were the Dokshukinski and Efremovski plants. The center of the ABE fermentation industry was situated at the Dokshukinski plant, where a central research institute was established. In 1962, an experimental continuous fermentation process was developed at the Dokshukinski factory. The process was then scaled up to provide a cascade process consisting of banks of up to eight 255,000 L and 275,000 L fermenters run in sequence. The semi-continuous battery fermentation process increased productivity by around 30% and was subsequently installed in the other Russian plants. The continuous cascade process utilized mixed substrates that typically consisted of a component of beet molasses making up between 50 and 70% (*wt/vol*) of the mash along with a component of starch such as wheat or rye flour. The technology for using lignocellulosic wastes as a feedstock was also developed after WWII. The Dokshukinski plant used mashes consisting of flour, a mixture of molasses, and hydrolysates of maize stubs and sunflower shells. In some plants, acid hydrolysates of maize cobs, hemp waste, sunflower shells, and other agricultural wastes were also added to the mash at a concentration of between 3 and 6%. The sludge from the continuous fermentation was either treated by thermophilic anaerobic digestion or mixed with yeast stillage to produce animal feed.

### 3.10. The Postwar Industrial Fermentation Process in Poland

Limited information is available regarding the development of the industrial ABE fermentation process in Poland, but quite an extensive industry existed after WWII. In 1937, there were numerous plants manufacturing industrial chemicals, and an ABE fermentation plant was begun at Chorzow, but it was not completed before WWII. After WWII, a Soviet-type command economy system involving state ownership of productive resources under a centralized administrative system controlled by the Communist Party was established in the Polish People’s Republic. From 1945 to 1989, the bulk of Polish agriculture remained in private ownership, but private enterprise only played a marginal role in the Polish economy. There are a few articles and reports by Logotkin and colleagues describing the method of processing sugar beet molasses in AB fermentation plants. The basic process was common to all the Polish plants. A culture reported as *C. acetobutylicum* was used to ferment a mash consisting of a mixture of beet molasses and meal that included barley flower, rye, or wheat meal with a ratio of 6.5 tons of molasses to 0.86 tons of flour. The fermenters had a capacity ranging from 75,000 L to 95,000 L, and the fermentation took between 48 and 60 h at 37–39 °C. Solvents were produced in the ratio of approximately 66% (*wt/vol*) butanol, 32% acetone, and 2% ethanol. The Polish process was said to also be used in the USSR. No information could be found with respect to the origin of the strains used or the actual number or location of the plants. Research was conducted in the 1960s and 1970s regarding the possibility of the use of agricultural wastes instead of food products, including the fermentation of molasses with the addition of bran and hydrolyzed cornstalks. The development of continuous culture was also investigated.

### 3.11. The Fermentation Process Developed in Czechoslovakia

Industrial production of ABE was also carried out in Czechoslovakia after WWII. An experimental plant was operated by the Ministry of Food Industry of the Czechoslovak Socialist Republic from 1952 until 1965. Research on ABE fermentation was initially started during WWII. Bacterial cultures were isolated mainly from soil and were selected and tested by Professor J Dyr, head of the Department of Fermentation Technology of the Institute of Chemical Technology in Prague. The various bacterial cultures were all classified as *C. acetobutylicum* and were specifically developed for use with the main crops to increase flexibility. The experimental plant was built at a distillery in Rájec-Jestřebí in Moravia, and initially, the main raw material was potatoes, but this was later changed to rye. The initial starch concentration ranged from 4.5–5% (*wt/vol*), and no other nutrients were added to the fermentation broth. Although sugar beet was extensively grown in the Czech Republic, efforts by the plant management to replace starch-based raw materials with beet molasses were refused because the stillage was mainly used as a supplement to animal feed and would have been of poorer quality. The plant had six 30,000 L and two 60,000 L fermenters. The process was run as batch fermentation with an average fermentation time of 36–38 h. The concentration of total solvents in the broth varied around 17–18 g/L. Annual production of solvents increased from year to year but never exceeded 1000 tons. The gases were not captured, and the stillage was used as cattle feed. Research on a continuous process was carried out but never tested on a large scale. The reasons for the closure of the plant in 1965 were the increase in feedstock price together with a decrease in the price of solvents. The equipment from the plant was cut up and sold as scrap.

### 3.12. The Industrial Fermentation Process Operated in Egypt

The sugarcane and sugar beet industry in Egypt has a long history, and several independent sugar mills were built in the late 1800s and early 1900s. In 1881, a sugar refinery was established at Hawamdia in Giza Governorate near Cairo under the name The Egyptian Refinery Company, with the purpose of refining the raw sugar necessary for local consumption instead of sending it to Europe for refining. In Egypt, most large-scale manufacturing establishments were nationalized beginning in the 1950s, and the Egyptian sugar industry was fully nationalized in 1961. The Egyptian Sugar & Integrated Industries company was established in 1956 as a state-owned enterprise, and the company later became one of the affiliated companies to the holding company for the Food Industries Group. The company specialized in the manufacturing and exporting of sugar from sugar cane and its by-products, including alcohol, molasses, yeast, and the manufacture of paper from sugar cane bagasse. Distillation factories of the Sugar and Integrated Company at El-Hawamdia Giza produced alcohol and organic solvents. This operated under a single entity called the Egyptian Sugar and Distillation Company. At this time, the Egyptian sugar industry produced around 20,000 tons of molasses per annum, of which about one-third was used by the fermentation industry. A long-term trade and aid agreement was reached with the Soviet Union in 1964, and it is reported that Egyptian solvent production was based on Soviet technology, and the process existed until 2008. The company operated a conventional commercial batch ABE fermentation process using Egyptian blackstrap molasses with a sugar content of up to 7% (*wt/vol*). There is an indication that the clostridial strains were isolated and developed in Egypt. A National Research Centre was established in Giza, Cairo, associated with the Dokki Factory. Several reports were published in the mid-1970s and 1980s on studies mainly undertaken on the optimization of nutrients on different types of Egyptian molasses, with some work also performed on the use of millet and other starch-based substrates.

### 3.13. The Initial and Later Industrial Fermentation Processes Established in Brazil

Brazil has been in the privileged position of being the largest producer of sugar in the world. Sucrose, which corresponds to one-third of the sugarcane biomass, is used to produce sugar and ethanol and contributes a major component to its agricultural and industrial wealth. A sugar mill, Usina Paraíso, was first established in Campos in the current district of Tocos, the sugar-growing region in the State of Rio de Janeiro. In 1900, the mill was acquired by a French company, Sucrerie du Cupim, along with several other plants, and the company was renamed Societé Sucreries Bresilienne in 1907. A French engineer, Victor Sence, was employed at the plant from 1900 to 1912 before founding his own company, the Usina Conceição in Conceição de Macabu, in 1913. Until its closure in 1993, this was one of the largest sugar-alcohol industries in the North of Rio de Janeiro. After WWII, an ABE fermentation process was established at the plant. The French company Ursines de Melle was involved in the design of the plant that was managed by a reclusive French fermentation technologist who isolated and developed solvent-producing strains of clostridia from Brazil. The factory used both blackstrap molasses and cane juice as the fermentation substrate. The plant consisted of a bank of 42 fermenters, each with a working volume of around 50,000 L, and employed batch sterilization, batch fermentation, and batch distillation technology. The commercial ABE process was the only producer of acetone, butanol, and butyl acetate from sugar cane in Latin America and operated from around 1948 to 1993 until the Usina Conceição ceased operations.

After operating uninterruptedly for more than 50 years, the Societé Sucrerries Bresilienne closed its activities in Brazil and sold its plants. Geraldo Silveira Coutinho bought Usina Paraíso in 1964 and managed the company together with his family. Seeking to diversify its activities, the group set up HC Sucroquímica in 2006 to produce acetone and butanol from sugarcane. They acquired the strains and technological information that had been used at the Usina Conceição plant. Four surviving strains were characterized as *C. saccharobutylicum* based on their phenotypic characteristics. The plant used a continuous sterilization process for sugarcane juice drawn directly from the sugar milling process with a sugar concentration between 6 and 7% (*wt/vol*). A bank of eight 350,000 L fermenters with a working volume of around 300,000 L were coupled with 200 L and 2000 L pre-fermenters. The fermentation time was 36–40 h with total solvents ranging from 16–22 g/L with a production of roughly 2000 tons p.a. Gases were discharged directly to the atmosphere, and the stillage was used as cattle feed. In 2012, the company experienced a devastating financial crisis during a general economic downturn that contributed to the bankruptcy of dozens of local sugar mills. The company had loans totaling more than BRL 4 million and was prosecuted for tax evasion and labor payment rights. As a result, the company was auctioned off by the Federal Justice in 2014 with an initial asking price of BRL 88 million, but it was not sold and was then placed under judicial recovery. The sugar mill was closed, and sugar cane from the estate was processed by a cooperative of producers. The company remains involved in legal disputes.

### 3.14. The Postwar Industrial Fermentation Processes Developed and Operated in China 

The industrial ABE fermentation process was first developed in China in the 1950s. The first commercial process was located at the solvent plant in Shanghai. Subsequently, the fermentation industry went through a period of rapid expansion, and by the early 1960s, ABE plants were operating in Shanghai, Beijing, and Wuxi, each with an annual production capacity of around 10,000 tons of solvents. During the period from 1965 to 1970, further expansion occurred, with ten more plants being erected at various locations with a total production capacity of 40,000 to 50,000 tons. Further expansion of the fermentation process resulted in around 30 plants being established, including those in Shandong and Hebei in the northeastern provinces that utilized starched-based substrates and a number of plants in the southern Guangxi province using molasses as the substrate. These new plants had an annual output of between 5000 and 10,000 tons of solvent, with the total annual production reaching around 170,000 tons.

A central institute of microbiology was established in Shanghai to serve the needs of the industry. Various strains of *C. acetobutylicum* were isolated and developed locally for use on starch-based substrates, including maize, cassava, potatoes, and sweet potatoes, to produce high yields. Special strains of *C. butanolacetonicus* were developed for the fermentation of molasses, and hyper-butanol-producing mutants were developed, giving an increase in butanol ratio of up to 70% (*vol/vol*) of total solvents. The initial process technology used was the conventional Weizmann-type batch process using *C. acetobutylicum*. In 1958, a multistage continuous process was developed at the Shanghai plant. The pilot process consisted of serial 100-ton fermenters, using 8% (*wt/vol*) maize mash as the substrate, operating on a fermentation cycle of 20 h. This was successfully scaled up to 200-ton fermenters, and the new technology was then transferred to plants all over China. The main problem reported with the multiple-stage process was bacterial contamination and culture degeneration. The problem of degeneration was largely overcome by the incorporation of stillage to enhance the nutrient composition of the mash. Around the beginning of the 1980s, the established ABE fermentation industry in China underwent another significant change when the decision was made to move many of the plants located in the big cities to agricultural districts. This plan also included a shift of plants from the southern region to the northern region to provide more effective access to grain-based substrates. The ABE process in China survived because of the availability of cheap grain. However, the fermentation process became less and less competitive with petrochemical synthesis, and progressively, by the end of the 20th century, almost the entire fermentation industry had been shut down, and the plants closed. In 2000, the demand for solvents had exceeded 280,000 tons, requiring 167,000 tons of solvents to be imported. By 2005, the demand had increased to 520,000 tons, but at this time, 500,000 tons were being produced synthetically in China.

## 4. The Resurgence of Interest in Biobutanol Production

Although the industrial production of both ethanol and butanol by fermentation preceded their production from petrochemicals, the latter quickly became the major supply of chemical feedstock. The fermentation route continued as an option, but the cost of the agricultural-based raw materials meant that it remained only marginally commercially competitive with oil-based chemical production. There are only a few countries with the capacity to produce the large volumes of agricultural-based raw materials required at competitive prices that have also had the appropriate technological capabilities. This capability has essentially been limited to the US with a large maize-based agricultural capacity, Brazil with a sugar-based agricultural capacity, and China with a mixed agriculturally based capacity. Currently, about 36% of all maize produced in the US is consumed by the ethanol industry, which supplies fuel blenders. The use of lignocellulosic biomass as a raw material could provide an alternative, but the cost of pre-treatment means that it has remained out of contention. Where the production of chemicals by the fermentation route has been employed by countries for strategic purposes rather than commercial profit, these constraints are of less importance. This is exemplified by the strategic production of solvents during the two world wars and the peacetime production in Russia and China. The resurgence of interest in the production of biobutanol in France in the 1980s is another example.

### 4.1. The National Program in France in the 1980s

At the beginning of the 1980s, France embarked on a strategic national program to replace 10% of its petroleum requirements with alternative fuels by 1990. This initiative resulted in several French institutions and firms exploring technologies to make gasoline substitutes. One of the major problems of adding methanol or ethanol to gasoline is the tendency for components to separate in the presence of water. This separation can be circumvented by the addition of solvents. Butanol, acetone, and iso-propanol were identified as the most suitable solvents for this purpose, with the advantage that they could be produced by fermentation. The Institut Français du Pétrole (IFP) was largely responsible for overseeing the coordination, devolvement, and progress of the program. The two agricultural crops most suitable for use in France were identified as Jerusalem artichoke and sugar beet. The research and development of an industrial ABE fermentation process was largely undertaken by the IFP. The IFP 903 strain (ATCC 39057) that was isolated and used for fermenting Jerusalem artichoke liquor was identified as the most promising candidate for this purpose. Research was undertaken to improve performance that led to the number eight mutant strain IFP 904 (ATCC 39058) that produced a total of 15 g/L of butanol and 10 g/L of acetone under optimal laboratory conditions. Both these strains were patented (US 4757010) as *C. acetobutylicum* but have subsequently been re-classified as *C. beijerinckii*.

A second aspect of the French program focused on the use of lignocellulosic substrates, including maize stover, maize cobs, cereal straw, and other types of biomasses. In 1984, the IFP and its French engineering subsidiary Technip began work on a pilot facility at Soustons to evaluate the industrial potential of the production of solvents from lignocellulose. The project included the building of a demonstration plant for conversion of biomass, designed by SunOpta, that came on stream in 1986. The plant included a 30,000 L fermenter for the production of the cellulase, two 25,000 L vessels for biomass saccharification, and a 50,000 L reactor tank. Batch ABE fermentations were performed on maize cobs, and a financial evaluation of the process showed that the economics of the process was strongly dependent on the market value of the by-product lignin. A slump in the price of crude oil led to the abandonment of the program, and the plant was sold to SAF-ISIS, Soustons, a French biotechnology company.

### 4.2. The Resurgence of Biobutanol Production in China

In this century, the most successful attempt to bring about a resurgence in the re-commercialization of biobutanol production has been in the People’s Republic of China. In the early part of this century, it was said that there were at least 16 plants in the country that were either functioning or being developed using maize or cassava starch as the major substrates. Most of the Chinese plants operate in a semi-continuous fashion, with each fermentation lasting up to 21 days. The plants typically house several trains of up to eight fermentation tanks with volumes of 300,000–400,000 L that are linked together in series. Conventional distillation is then used to recover the solvents. Most plants are located next to ethanol plants to reduce utility and operating costs and tend to share effluent treatment facilities. Biogas produced by anaerobic digestion is used to generate heat and power. Most plants do not utilize the gases from the fermentation.

Prior to 2008, over USD 200 million was invested in China to install 210,000 tons p.a. of solvent capacity with plans to expand this to a million tons p.a. There were six major plants that produced about 30,000 tons of butanol p.a. from starch. The largest of these were the Anhui COFCO Biochemical Company, the Jinyuan Alcohol Industry Company, and Cathay Industrial Biotech based in Shanghai. Cathay completed the construction of its fermentation facility at its biorefinery in Jilin Province in 2008 and scaled up its production capacity to 21 million gallons per year of biobutanol in 2009 using maize-based feedstock. The company also started using agricultural residual wastes, including maize cob and maize stover, as feedstock in 2011. After the oil price crashed in 2008, the butanol price fell from USD 2400 to USD 1000 per ton, and the Chinese plants ceased production. By the end of 2010, the butanol price had recovered, and several plants restarted production. More recently, China’s biofuel policies have continued to wane as the priority for the People’s Republic of China, and investment in the sector has declined.

### 4.3. Industrial Biobutanol Production Initiatives in the West

Due to a revival of the interest in biobutanol production in the early part of the 21st century, several fuel and biotechnological companies and scientific organizations focused their efforts on investigating the physiology, genetics, and metabolism of butanol-producing clostridia. The companies that have been involved in developing technology for biobutanol manufacture have primarily been biotechnology companies, some of them start-ups with opportunistic capital or spin-offs from academic research. There has also been intense activity to merge companies or acquisitions by the larger companies. Some of the main companies involved are reviewed below.

Green Biologics Ltd. (GBL) began as an industrial biotechnology company headquartered in the UK, founded by Edward Green in 2003. The company spent the first decade developing a technology based on clostridia for the re-commercialization of solvent production. In 2011, GBL merged with Butylfuel, a US-based biobutanol technology company founded in 1991 by David Ramey, providing a pilot scale plant at its facilities in Columbus, Ohio. The company then built a demonstration-scale facility in Emmetsburg, Iowa, with the aim of retrofitting uneconomic US ethanol plants to produce solvents. GBL raised roughly USD 100 million in equity financing from angel investors and venture capital firms for this purpose. In 2014, GBL purchased the Central MN Ethanol Corn distillery in Little Falls, Minnesota, with the intention of repurposing the plant, which included a 21 million gallon per year ethanol production unit, to produce butanol and acetone utilizing its proprietary advanced fermentation process technology platform. The acquisition was made through Central MN Renewables, an affiliate of Green Biologics Inc. Construction began in late 2015 to retrofit the facility, and the plant began operations in 2016. In 2018, the company secured contracts for several renewable products, including barbecue lighter fluid and biobased acetone nail polish remover. In 2019, the company began to run out of money, and the GBL board concluded that it was not possible to secure the funding necessary to enable it to continue with plans to build up sales and production levels to the point of cash break-even. As a result, GBL had no option but to close the plant, and GBL ceased operations in July 2019. One of the problems the company faced with small-volume fermentation-based production was the ability to compete with fossil-fuel production on both cost and volume. There were also issues associated with retrofitting a facility designed for ethanol production rather than a custom-built facility where contamination could more easily be controlled.

Cobalt Technologies was founded in 2005 to develop technologies to produce biobutanol using various cellulosic feedstocks. In 2010, the company demonstrated biobutanol production from beetle-killed lodgepole pine trees and, in 2011, raised venture funding with the intention of supplying biobased jet fuel to the US Navy. The following year, Cobalt Technologies established a partnership with RHODIA and Bunge to operate a demonstration facility using sugarcane bagasse in Brazil. In 2015, Cobalt Technologies went bankrupt, and assets were liquidated. The Eastman Chemical Co. ventured into biobutanol production in 2011 when it acquired TetraVitae Biosciences Inc. and its assets. More recently, Celtic Renewables Limited, a company based in Edinburgh, embarked on the construction of a demonstration plant to produce biobutanol from the whisky industry by-products of draff and pot ale that is scheduled to come on stream.

Two other companies, Gevo, founded in 2005, and Butamax, founded in 2009 as a 50/50 joint venture between BP and DuPont, developed their own proprietary microorganism to convert fermentable sugars into isobutanol through synthetic biology with Gevo’s proprietary integrated fermentation technology platform using genetically modified *Escherichia coli* with Butamax technology based on engineered *Pseudomonas*. In 2010, Gevo purchased an ethanol plant in Luverne, Minn., with the intent of converting it to isobutanol production. Butamax built a demonstration plant at Hull in the UK in 2010 and established a research facility in Brazil in 2011. Butamax achieved victory in Gevo’s lawsuit for copyright infringement in 2014, and the following year, they entered into patent cross-license and settlement agreements. In 2017, Butamax acquired the Nesika Energy ethanol facility in Scandia, Kansas, with the intention of adding bio-isobutanol capacity to the facility. Butamax subsequently went through reorganization following DuPont’s merger with Dow and is now a subsidiary of BP and Corteva Agriscience. Neither company has as yet produced significant amounts of bio-isobutanol. Other companies involved have focused on the metabolic engineering of the clostridial strains to optimize the yield of n-butanol. Phytonix proposes to make butanol from waste carbon dioxide using cyanobacteria, and Abengoa intends to use the catalytic condensation of bioethanol to produce biobutanol.

Brazil is considered one of the few countries with the capability of a consistent production of n-butanol. The companies involved in developing technology for biobutanol have primarily been biotechnology companies, some of them start-ups with venture capital or spin-offs from academic research. There has been intense market activity to merge companies or acquisitions from larger companies. In 2014, several companies, which included Amyris, BASF, BioChemtex, BP, Centro de Tecnologia Canavieira, Dow Chemical, DSM, DuPont, GranBio, Novozymes, Raizen, and Rhodia, formed the Brazilian Industrial Biotechnology Association with the objective of fostering industrial biotechnology in Brazil. A number of plants have been planned and designed, but HC Sucroquímica operated the fermentation plant at Usina Paradíso, with an estimated output between 2000–10,000 tons p.a. from 2006 until 2014, was one of the few companies that has achieved a consistent production of n-butanol using sugar cane juice as a fermentation substrate.

### 4.4. The Resurgence of Interest in Lignocellulose-Based Biobutanol

Various companies, including Tetra Vitae Bioscience and Arbor Fuel Inc. in the US, Butalco in Switzerland, and Metabolic Explorer in France, have been involved in attempts to develop processes for biobutanol production from lignocellulose. At Dartmouth, Professor Lee Lynd has been involved in the utilization of plant biomass for the production of solvents and is the co-founder of Terragia Corp, Mascoma, and Enchi Corps. Technology for biobutanol production by the fermentation of wood hydrolysate was also explored by the Aalto University in Finland and Russia. Large-scale biobutanol production from wood hydrolysates was planned at the Tulunskii hydrolysis factory in Russia based on the technology developed by ZAO Biosintezbelok.

A more attractive option is the use of the pentose-rich hemi-cellulose streams resulting from agricultural, paper pulp, and wood processing plants. Typically, this fraction does not require hydrolysis but still usually needs some detoxification to reduce growth inhibitors. This approach has been demonstrated in China by Songyuan Laihe Chemicals. A 600 tons p.a. pilot plant was built to ferment sugars contained within the hemi-cellulose fraction from pre-treated maize stover.

### 4.5. Future Prospects

There are several commercial organizations that specialize in providing reports on biofuel market research, global industry trends, and opportunities. A number of these firms give market forecasts that vary widely but estimate that the biobutanol market size could reach up to USD 20 billion by around 2030 based on the increasing demands for environmentally bio-based products. These reports state that there is a high demand for biobutanol but tend to lack specific information as to where and when the large volumes required will become available. Currently, the short answer is unclear and not easy to predict. From past history, the interest and provision of funding for biobutanol production have tended to only be forthcoming during times when the supply of oil has been threatened, and prices have skyrocketed. Once the fuel crisis is over, both the interest and the funding dry up, and much of the progress that has been made is lost until the next cycle occurs.

## 5. Features of the Solvent-Producing Clostridia

The clostridia are a large, diverse group of anaerobic bacteria that includes mesophiles and thermophiles. Members of the genus *Clostridium* were originally classified as rod-shaped, endospore-forming, obligate anaerobic bacteria, most with Gram-positive cell wall structure. The numerous species inhabit a wide range of anaerobic environments, including soils and the intestinal tract of animals and humans. The group includes a number of significant and important human and animal pathogens, and the clostridia are known to produce the greatest variety of toxins of any bacterial class.

### 5.1. The Genus Clostridium

The clostridia belong to the Bacillota (synonymous with *Firmicutes)* phylum of bacteria. The initial definition of the genus *Clostridium* encompassed a very large and diverse group of species that were placed in the unnatural order defined as *Clostridiales*, which included the class *Clostridia.* Comprehensive molecular taxonomic and evolutionary analysis has shown that the original definition is inconsistent with a definition for a genus and has confirmed that the clostridia are not a monophyletic group. Instead, the clostridia now include multiple genera, which highlights the importance of phylogenomics for taxonomic studies. This revision led to the need to redefine the clostridial group taxonomically and has resulted in the classification being revisited several times, with many new genera being split off [14,15,16,17,18,19]. In 1994, Collins et al. [14] split the traditional genus into twenty clusters, with cluster I containing the type species and close relatives of most of the well-known *Clostridium* species that now belong to “cluster I” (*sensu stricto*). The other clusters are no longer classified as *Clostridium*. As of October 2022, there are 164 recognized and validly published species in the *Clostridium* genus (*sensu stricto*). Examples of other clostridial clusters are the *IV* and *XIVa* clusters that efficiently ferment plant polysaccharides composed of dietary fiber, making them important and abundant taxa in the gut and rumen of animals and the human large intestine. This reclassification includes *Clostridium difficile,* an important pathogen that now belongs to a distantly related genus reclassified as *Clostridioides difficile*. Other examples include the extensively studied cellulolytic species with applications in biotechnology that include *Clostridium thermocellum,* now reclassified as *Ruminiclostridium thermocellum*, and *Clostridium phytofermentans* now reclassified as *Lachnoclostridium phytofermentans* [14,15,16,17,18,19].

Originally, the naming and status of each species of *Clostridium* was determined using a wide diversity of phenotypic features. The International Code of Nomenclature of Bacteria was instituted to determine the correct name for each described species, and this is the name that must be adopted under these rules. Bergey’s Manual of Determinative Bacteriology and the companion volume Index Bergeyana was first established in the US in 1923 to assess and determine the legitimacy of the names of members of bacterial taxa, and the final ninth edition was published in 1993. The use of phenotypic characteristics alone for taxonomy imposed serious limitations, and continual revision was required to update the Index. The manual was intended to assist in the identification of bacteria, and no attempt was made to establish a complete hierarchical classification system as it was recognized that this was not possible. The manual was envisioned to stimulate further work on bacterial taxonomy and classification, but it was acknowledged that any classification reflecting the phylogenetic relationships of bacterial species and strains would have to be based on genotypic characteristics rather than phenotypic characteristics [20].

### 5.2. Solvent-Producing Clostridium Species

Louis Pasteur was the first microbiologist to observe that certain anaerobic spore-forming bacteria could produce butanol (butyl alcohol). During the late eighteenth century, several early European microbiologists isolated and described strains of anaerobic endospore-forming bacilli that produced butanol that were later assigned to the genus *Clostridium.* Acetone production was first reported by Schardinger in 1905. The ability to produce butanol appears to be almost exclusively confined to mesophilic species belonging to the genus *Clostridium* (*sensu stricto*) and occurs in species belonging to different taxonomic clades. In addition to acetic and butyric acid, the solvent-producing clostridia also produces butanol, acetone/iso-propanol, and ethanol by the fermentation of a wide spectrum of carbohydrates. A common feature is their ability to utilize simple as well as complex sugars, including pentoses and hexoses, through glycolysis and the nonoxidative pentose phosphate pathways. Many species are able to degrade complex polysaccharides into simple monosaccharides due to their ability to secrete large quantities of extracellular enzymes.

Among the first species to be intensively studied was *C. acetobutylicum,* isolated and developed by Chaim Weizmann for the production of acetone during WWI. Industrial strains belonging to this species have been used mainly for solvent production from starch-based substrates. The ability to produce butanol has also been reported in the closely related species *C. aurantibutyricum*, *C. felcineum*, and *C. roseum*. These are pectinolytic bacteria that have been used in retting and are usually orange or pink-pigmented. Molecular taxonomic studies indicate that they should be classified as a single species. The type species of *C. aurantibutyricum* has been reported to produce both acetone and iso-propanol in addition to butanol. *C. pasteurianum* is another species in the same clade as *C. acetobutylicum* that can produce limited amounts of butanol from glycerol. The type species of *C. pasteurianum* has been extensively studied due to its ability to fix nitrogen.

A second clade includes three closely related saccharolytic species *C. beijerinckii, C. saccharobutylicum*, and *C. saccharoperbutylacetonicum,* that have been widely used for the industrial production of solvents. In addition to these species, there are a number of other clostridial species reported to produce varying amounts of butanol that have never been used in industry. The *C. tetanomorphum* and *C. puniceum* species produce butanol in equimolar amounts with acetone and ethanol as end products. Other species, including *C. ljungdahli*, *C. sporogenes*, and *C. cadaveris*, produce insignificant amounts of butanol. *Thermoanaerobacterium thermosaccharolyticum*, previously classified as *C. thermosaccharolyticum,* is also reported to produce small amounts of butanol. Under high substrate loadings (100 g/L), *C. thermocellum* can produce isobutanol and other overflow metabolites.

### 5.3. Common Features of Industrial Solvent-Producing Strains

Of the mesophilic, saccharolytic clostridial species known to produce solvents, only a few are capable of producing butanol with sufficiently high fermentation yields to be commercially viable. This ability is confined to industrial strains of *C. acetobutylicum, C. beijerinckii, C. saccharobutylicum,* and *C. saccharoperbutylacetonicum*. Although these strains do differ phylogenetically, they all exhibit similar characteristics and share features in common.

### 5.4. Substrate Utilization 

The traditional raw materials utilized for the industrial process are either starch-based substrates that include maize, cassava, sweet potatoes, potatoes, and other cereals or sugar-based substrates that include sugarcane juice, blackstrap molasses, high test molasses, and sugar beet molasses. The use of simple mono- and disaccharides simplifies the fermentation process and significantly decreases the cost of biosynthesis, and all four species are effective at utilizing starch-based substrates without requiring initial hydrolysis. Successful commercial fermentations using molasses were never achieved using *C. acetobutylicum* strains. Although sugars and starch are good substrates for butanol production, their use is mitigated because of their high price. The utilization of these crops for the production of solvents is also controversial due to competition for their use as food. Solvents can be produced from other types of biomass containing mono-, oligo- and polysaccharides such as Jerusalem artichokes and whey permeate. Other types of agricultural and industrial by-products, wastes, and residues also offer potentially economically viable options for production.

Lignocellulosic biomass is a potentially attractive feedstock due to its year-round availability and lower costs of harvesting and transporting in comparison to crop biomass. Lignocellulosic hydrolysates have been used but require significant upstream processing that usually consists of pre-treatment, hydrolysis, and detoxification. Promising abundant sources of lignocellulosic biomass include bagasse, maize cobs, maize stover, wheat straw, peanut shells, alfalfa stems, flax shives, hemp hurds, and other food processing wastes. The use of dedicated crops such as switchgrass, salix, spruce, and poplar has also been investigated. However, lignocellulosic biomass cannot be converted directly, and additional expensive physical and chemical processing is required. The pre-treatment method depends on the type of biomass used, but the most common methods are dilute acid pre-treatment and steam explosion pre-treatment. Enzymatic hydrolysis can provide a more effective and environmentally friendly option; however, the expense substantially increases the cost of saccharification.

### 5.5. The Two-Phase Feature of the Industrial Batch Fermentation Process

The industrial ABE fermentation process has typically been operated as a batch process with only limited modifications. The standard batch fermentation occurs in two stages. During the initial acidogenesis phase, the inoculum grows exponentially, and sugar is converted into mainly acetic and butyric acid along with CO_2_ and H_2_, resulting in a decrease in pH. Towards the end of acidogenesis, the rate of H^2^ production falls as the cells shift metabolic activity from acidogenesis to solventogenesis in response to the low pH. The concentration of the undissociated butyric acid triggers the phase change from the acidogenic to the solventogenic phase, and the phase shift is accompanied by the cessation of growth and the initiation of sporulation. Normal cell division is inhibited, and forespore septation occurs at one or occasionally both poles of the cell. This is usually accompanied by the accumulation of granulose, creating swollen clostridial forms. Solvent production and sporulation are interlinked under the control of the *spo0A* gene. In the second phase, acetate and butyrate act as co-substrates for the synthesis of butanol, acetone, and ethanol. Most of the acids are consumed during the solventogenic phase, but some acids usually remain in the final fermentation product. At the end of the solventogenic phase, the concentration of butanol reaches a level that inhibits bacterial metabolism and the formation of mature spores. The fermentation process usually takes between 30 and 75 h to reach completion, depending on the strain and process used. The time involved in the batch process is extended by the need for additional time-consuming steps of transfer for distillation, cleaning, sterilization, refilling, and re-inoculation of the bioreactor. With anaerobic fermentation, the size of the fermenter unit is not limited by oxygen or heat transfer rates, so banks of very large volume unstirred bioreactors can be used for production.

### 5.6. Continuous Fermentation

In theory, the limitations of batch fermentation can be avoided by using continuous fermentation processes. However, the nature of the two-phase fermentation makes this very challenging to achieve. The most common strategies for continuous fermentation that have been used are free-cell systems and immobilized-cell systems. Cell recycling in the free cell fermentation process usually employs mechanical or air-lift agitation. Cell immobilization has been developed using different bioreactor systems, including fibrous bed and packed bed bioreactors. Semi-continuous processes have been developed that utilize cascades of fermenters.

### 5.7. Butanol Toxicity and Tolerance

The limited tolerance to butanol of around 15 g/L is a key aspect affecting the economics of the industrial ABE fermentation. The limitation of low concentrations of butanol in the fermentation broth results in high product recovery costs. Butanol toxicity is mainly caused by its hydrophobic nature, which causes an increase in the fluidity of the cell membrane. This affects the function of the cell membrane as the regulatory barrier between the cell interior and exterior and the trans-membrane pH gradient is destroyed, resulting in energy limitation. Various techniques for the removal of butanol have been investigated, including cell recycling coupled with membrane filtration, liquid–liquid extraction, gas stripping, adsorption, reverse osmosis, perstraction, and pervaporation. Other approaches that have been investigated have used a variety of strategies to isolate butanol-tolerant strains of solventogenic clostridia. These include targeted genetic modifications such as the overexpression of genes encoding heat-shock proteins, modification of fatty acids synthesis, serial transfer, and adaptation, as well as random mutagenesis. In some studies, the strains that exhibited higher tolerance and improved survival in the presence of butanol also exhibited higher butanol production. However, other studies reported that increased butanol tolerance did not result in an increase in butanol production. These findings suggest that there could be a strain-specific connection between butanol tolerance and production in some instances, while in other cases, no direct connection exists between the two mechanisms.

### 5.8. Limitations

The substrate and energy costs associated with producing solvents by fermentation have tended to make the fermentation route commercially uncompetitive with petrochemical processes. When compared to the ethanol fermentation process, the main limitations of the ABE fermentation process are the solvent yield of around 2% and a batch fermentation consisting of an acidogenic phase followed by a solventogenic phase. The need to separate a low concentration of mixed solvents adds to the complexity and cost of distillation and the two-phase nature of the fermentation, and the requirement to maintain anaerobic conditions, coupled with a much greater susceptibility to contamination makes the operation of the fermentation process much more demanding. Unless the industrial process can be run day in, day out, week in, and week out without failures, it will struggle to remain commercially viable.

## 6. Molecular Taxonomy and Genome Sequencing of Solventogenic Clostridia

### 6.1. The Importance of the McCoy Culture Collection

It is largely due to the foresight of Elizabeth McCoy (Figure 1) that a significant number of strains have survived. McCoy was a pioneer of 20th-century industrial bacteriology who graduated from the University of Wisconsin in 1925. She then undertook graduate studies with E.B. Fred, focused on bacteria that can produce acetone and butanol. Later, she made a major contribution to understanding nitrogen-fixing bacteria, and during WWII, she contributed to the US war effort by scaling up penicillin production and went on to discover the antibiotic oligomycin. During her career, McCoy trained a total of 47 PhD and 110 master’s students, many of whom followed in her footsteps and went on to distinguished careers of their own.

Through a close working relationship with the Commercial Solvent corporation, McCoy isolated and patented the first successful industrial clostridial strain used for the production of solvents from molasses. She also acted as an expert witness for the company in their successful legal actions taken against the Synthetic Products Company in the UK and against the Union Solvent Corporation in the US. As part of these proceedings, she accumulated an extensive collection of reference strains.

The original McCoy Strain Collection consists of 159 strains maintained as spore stocks. Her A strain collection includes many strains she obtained from other microbiologists working in the field, as well as the strains that she and her students isolated. She later patented her A16 strain as *C. madisonii.* Her B strain collection consisted of patented and reference trial strains, some of which were duplicates of the A series. When she retired, the strain collection went to Leland McClung at Indiana University, and when he retired, strains were passed on to Jian-Shin Chen at the Virginia Polytechnic Institute. The spore stocks of 127 strains are now held by Steven Stoddard (microberesearch.com). A list of some of the most important strains from her collection, giving their origins and distribution, is provided in Table 1.

### 6.2. Patenting of Industrial Solvent-Producing Clostridia 

From the early 1900s, the importance of patenting industrial solvent-producing clostridial strains was recognized, and amongst the first strains to be patented were the Fernbach and the Weizmann strains. Taking out a patent to cover a particular industrial fermentation process and the usage of a particular industrial strain provides legal protection for a period of 20 years. The patenting of a strain required a full description of the original method of isolation and its phenotypic characteristics, as well as its fermentation performance and characteristics. These industrial strains remained closely guarded, but the patent application provided valuable information that was used by other parties seeking to isolate and develop their own strains for industrial use. Useful lists of the main industrial patents were provided by Beesch [9] and Jones and Keis [21]. Once the companies to which the patents had been ceded stopped operating or were no longer interested in the possibility of licensing their strains, most of these industrial strains were lost.

### 6.3. Solvent-Producing Strains Held in International Culture Collections

Many of the existing clostridial strains of industrial importance are held in international culture collections. The World Federation of Culture Collections directory lists a total of 834 microbiological culture collections. Many of the clostridial strains originate from the McCoy strain collection (Table 1), while other strains have been added from many other sources. Other solventogenic strains are held by commercial companies, research institutes, and universities.

The genomes of species and strains covered in this review were derived for the LanzaTech David Jones strain collection. The strains that were obtained from international culture collections are listed in a table (Appendix A). These strains were originally sourced from five culture collections. The American Type Culture Collection (ATCC) was founded in 1925 by a committee of scientists who recognized the need for a central collection of microorganisms that scientists worldwide could use to conduct research to advance the science of microbiology. The first ATCC catalog was published in 1927 and started charging for cultures to cover costs in 1930. The National Centre for Agricultural Utilization Research was one of four Regional Laboratories set up by the Agricultural Adjustments Act of 1939. Peoria, Illinois, was chosen to host this facility, named the Northern Regional Research Laboratory (NRRL) that maintains the Agricultural Research Service Culture Collection. In the UK, the National Collection of Industrial Bacteria was created in 1950 when the collection took over the non-pathogenic cultures held by the UK’s National Collection of Type Cultures. This collection was merged with the National Collections of Marine and Food Bacteria to create the culture collection of the National Collection of Industrial, Food, and Marine Bacteria, which is now curated by NCIMB Ltd. In Germany, the Leibniz Institute DSMZ-German Collection of Microorganisms and Cell Cultures was founded in 1969 as the national center for culture collection in Göttingen. It was originally part of the Gesellschaft für Strahlenforschung (GSF) and was later moved to Braunschweig to become an independent institution. In Japan, the Japan Collection of Microorganisms (JCM) was established by the Microbe Division in RIKEN-BCR in 1981.

### 6.4. The Taxonomy and Classification of Solvent-Producing Strains

Between WWI and WWII, a considerable amount of technical and scientific information regarding the industrial ABE fermentation process was published. After WWII, the use of the fermentation route for the commercial production of solvent declined. As a result, the amount of published material relating to the fermentation waned. From the 1970s onwards, there was a gradual increase in the number of groups in universities undertaking scientific research. At this time, clostridial strains that produced acetone and butanol were generally considered to be *C. acetobutylicum* strains, while those that produced iso-propanol were considered to be *C. beijerinckii* strains. Many research groups undertook studies on different strains, giving conflicting results. The confusion increased as a result of the NCIMB 8052 *C. acetobutylicum* type strain being mis-labeled and later shown to be a *C. beijerinckii* strain. It became apparent that a revision of the taxonomy was needed, and the availability of new molecular biological and genetic techniques provided the means to accomplish this.

### 6.5. Molecular Biological Phylogeny and Taxonomy Studies

The limitation of the taxonomy and nomenclature of the solventogenic clostridia based on phenotypic characteristics alone led to a number of taxonomic and phylogenetic studies utilizing molecular biological techniques, including DNA/DNA hybridization, rRNA sequencing, and genomic DNA restriction analysis [22,23,24,25,26]. As a result of these studies, it was possible to propose a revision of the phylogeny and taxonomy of the solventogenic clostridia. One of the most important discoveries was that the strains that were surveyed belonged to two separate phylogenetic clades that were only distantly related. One clade consisted of the original *C. acetobutylicum* strains that were utilized for the production of solvent from maize and other starch-based substrates. The other clade consisted of later strains that were utilized for the production of solvents from molasses and other sugar-based substrates. A number of these strains that had been classified as *C. acetobutylicum* were shown to be *C*. *beijerinckii* strains. These studies showed that the strains belonging to the saccharolytic clade consisted of two recognized species, *C. beijerinckii* and *C. saccharoperbutylacetonicum*. Strains belonging to a third species that had previously been unrecognized were classified as *C. saccharobutylicum* [24]. This revised classification was formally recognized and adopted and enabled the construction of a phylogenetic tree based on 16S rRNA sequences. An example of a tree showing these features is provided in Figure 2.

### 6.6. The David Jones Strain Collection

As a basis for studies undertaken on clostridial molecular biology, phylogeny, and taxonomy, a collection of solvent-producing clostridia and related strains was assembled to provide as broad a survey as possible. This strain collection, now referred to as the David Jones (DJ) collection, consisted of an original strain collection from the University of Cape Town. These were added to by additional production strains obtained from NCP after the company was sold off. The collection was expanded further by strains donated by a number of international culture collections for the purpose of the taxonomic study. Ultimately, the DJ collection consists of strains from essentially two different culture collections. One collection consists of NCP production strains that date back to 1944. These included examples of some of the earliest spore stocks sent from both CSC in the US and CS Ltd. in the UK, as well as later strains propagated by NCP. The collection was added to by strains from Japan donated by Professor Hongo. The other culture collection consisted of strains obtained from the international culture collections or derivatives of these strains provided by other research groups. After David Jones retired from the University of Otago, the culture collection was ceded to the LanzaTech company that had been established in New Zealand.

### 6.7. Genome Sequencing

The two solvent-producing species that have been most intensively studied are *C. acetobutylicum* ATCC 824 and *C. beijerinckii* NCIMB 8052. They have served as model systems for studying the physiology, metabolism, genetics, and systems biology of clostridia with industrial applications. These two strains were the first to have their genomes fully sequenced. The genome of *C. acetobutylicum* ATCC 824 was sequenced in 2001 [27], and the genome of *C. beijerinckii* NCIMB 8052 was sequenced by the DOE Joint Genome Institute in 2007. This enabled the genomes of the two solventogenic species to be compared at the gene level, leading to advancements in our understanding of the metabolic and regulatory networks and proteome. There are currently over 300 *Clostridium* genome sequences in the NIH GenBank database for species and strains that are known to produce acetone, butanol, ethanol, and iso-propanol, along with phylogenetically closely related strains.

### 6.8. The European Genome Sequencing Project

A joint genome sequencing project was established by two groups in Germany and four groups in the UK that was published in 2017 [28]. The genomes of a total of 30 saccharolytic *Clostridium* strains representative of the species *C. acetobutylicum, C. aurantibutyricum, C. beijerinckii, C. diolis, C. felsineum, C. pasteurianum, C. puniceum, C. roseum, C. saccharobutylicum,* and *C. saccharoperbutylacetonicum* were determined with 10 of the genomes sequenced completely. These genomes were compared to another 14 published genomes of other solvent-forming clostridia. Genome analysis confirmed the revised classification and phylogeny based on the earlier molecular biological studies and identified several misclassified species. The phylogeny of the strains was analyzed by multi-locus sequence analysis (MLSA) based on the detected core genome as well as Average Nucleotide Identity (ANI) analysis. The phylogenic tree that was constructed confirmed two main clades (I and II) with several subclades that correlated with the core pangenome analysis. This emphasized the separation of the solvent-producing clostridia into two major groups that are not closely related.

Genome sizes of the 44 genomes analyzed in this study varied between 4.099 Mb with about 4000 genes for *C. acetobutylicum* NCCB 24020 and 6.666 Mb for *C. saccharoperbutylacetonicum* N1-4 with 5937 genes was the largest genome in the solvent-producing clostridia. A whole-genome comparison based on protein-encoding genes revealed a core genome shared by all 44 strains of 547 ortho groups (OGs) with a pan-genome of 31,060 OGs. There was a broad range of genome-specific OGs that varied between 11 and 737 OG’s, which, with three exceptions, was smaller than the core genome of all 44 strains studied [28].

This project provided the basis for a comprehensive study of solvent production that underlined the differences in energy conservation mechanisms and substrate utilization between strains. The study enabled the direct comparison of selected industrial strains at the genetic level and the identification of possible engineering targets for improved solvent production. The long-term aim of the project was to support data mining to facilitate the development of more robust and sustainable fermentation routes to produce acetone and butanol for chemical and biofuel applications.

### 6.9. The Joint US LanzaTech JGI Genome Sequencing Project

LanzaTech is an industrial biotechnology company founded in New Zealand in 2005 and now based in Illinois in the US that has developed a gas fermentation platform for converting industrial emissions and wastes into valuable chemicals. It uses proprietary bio-reactors to convert CO_2_ to ethanol for making fuels and chemical production [29]. Initially, around 2007, the company screened the strain collection for acetogens and later *C. autoethanogenum* for the production of acetone and iso-propanol by gas fermentation at an industrial pilot scale. In collaboration with the US Department of Energy Joint Genome Institute (JGI), strains from the DJ culture collection of solventogenic clostridia had their genome sequences determined and analyzed. Using a combinatorial pathway library approach, the genomes were then mined for superior enzymes, the genes of which were used to engineer *C. autoethanogenum* to produce acetone and iso-propanol [29].

The strains from the DJ collection were allocated arbitrary codes from DJ001 to DJ350, and a total of 270 genomes were sequenced, consisting of 7 *C. acetobutylicum*, 194 *C. beijerinckii*, 5 *C. butyricum*, 57 *C. saccharobutylicum*, 4 *C. saccharoperbutylacetonicum*, and 3 *C. tetanomorphum* genomes [30].

Each of the major branches leading to species-level clades was confirmed, although some misclassifications were identified. The genomes were uploaded to GenBank, and a broader phylogenomic analysis of the genus was also conducted from genome assemblies in the GenBank database around the same time [30]. A phylogenetic tree showing the phylogenomic relationship of the six *Clostridium* species investigated in this study is shown in Figure 3.

The uploaded genomes of the DJ strains that were sequenced reflect their species status, but the DJ codes do not provide any information with respect to the strain origin or nomenclature. A table has been provided that enables the conversion of the DJ codes to the standard established nomenclature and origin for the strains within each species (Appendix A).

As part of the collaborative project with JGI, the genome sequences were analyzed, providing a wealth of new information and insights. This included a compendium of the key attributes of the newly sequenced genomes, such as genome size, G+C ratios, number of ORFs and genes, RNA species, etc. Core biosynthetic genes involved in metabolism and solvent production, as well as secondary metabolites and biosynthetic gene clusters, were analyzed, and a wide array of secondary metabolites, including antibiotics, lantipeptides, and bacteriocins, were identified. Prophage diversity and their evolutionary relationships, CRISPR-Cas diversity, and the occurrence of plasmids and insertion elements were also investigated [30]. The key genome attributes of the industrial species have been summarized in Table 2.

## 7. Phylogenomics and Phylogeny of the Solventogenic Clostridia

### 7.1. Comparative Phylogenomics of Clade 1 Species

The phylogenetic grouping of the solvent-producing species in Clade 1 is characterized by a common type I *sol* operon organization (gene order *adhE–ctfA–ctfB*) with a separate *adc* operon located adjacent and being transcribed convergently. A *pdc* gene encoding pyruvate decarboxylase is present in these species, and *rnf* genes involved in the generation of an additional ion gradient from reduced ferredoxin are absent. One major difference is that the *sol* operon in the *C. acetobutylicum* strain is located on a megaplasmid, whereas in the other species, it is located on the chromosome [28].

Genotypic comparisons have shown that the differences between *C. aurantibutyricum*, *C. felsineum*, and *C. roseum* species are marginal and not sufficient to justify separate species designations for these strains [28]. This requires that amended descriptions and a common species name be implemented for this group of pigmented pectolytic solvent-producing strains. *C. felsineum* was first described in 1917 and was recognized in Bergey’s Manual of Determinative Bacteriology in 1930 and 1934 and would have precedence for the name of the species with ATCC 1778, DSM 794, NCIMB 10690 as the Type strain. In addition to the genome of *C. pasteurianum* type strain ATCC 6013, there are a number of genome sequences, but all these strains are variants of the type strain. The strain currently classified as *C. pasteurianum* BC1 exhibits many unique characteristics, and it is clear that phylogenetically constitutes a new species and requires renaming [31].

There are now at least 14 genome sequences for *C. acetobutylicum,* including variants of the original Weizmann strain. Analysis revealed that all these strains are very closely related with ANI values of around 100%. The exception is the GXAS18_1 strain isolated in China. These strains were isolated in Europe, Britain, North America, South America, Japan, and China over a period of more than a hundred years. Many of the strains have been sub-cultured many thousands of times. The remarkable feature is that the genomes of all these disparate strains are all so similar. This indicates that the genomes are remarkably stable, highly conserved, and perfectly adapted to function in the environmental niche they occupy.

### 7.2. Comparative Phylogenomics of Clade 2 Species

The second clade consists of the most widely used industrial strains used mainly to produce solvents from inverted sugars and molasses. They include *C. saccharobutylicum*, *C. saccharoperbutylacetonicum*, *C. beijerinckii*, as well as *C. butyricum* and *C. puniceum*. There are a few other related strains that constitute separate species that require new descriptions and designations. The members of this clade encode the *sol* operon in the gene order *ald–ctfA–ctfB–adc* and lack a *pdc* gene [28].

Only two strains of *C. saccharoperbutylacetonicum* isolated and patented by Hongo and Nagata from the soil are available from international culture collections. These are the high butanol-producing N1-4 strain and the N1-504 strain. Two variants of the N1-4 strain were lodged with culture collections. The N1-4 strain is the type strain, and the N1-4 HMT strain was lodged as a lysogenic strain harboring the inducible HM-1 prophage. Later studies revealed the HM-1 phage has a very small genome and does not integrate into the host genome but can become incorporated into the endospore as a pseudo-lysogen. Over time, this strain has lost the phage and can no longer be induced. The N1-504 strain was originally considered to be a phage-resistant mutant of N1-4. Subsequent genetic studies have revealed that the strain differs markedly from the N1-4 strain such that it could be considered a separate subspecies. The NCP P262 strain is the type strain for *C. saccharobutylicum* and is currently the only strain of this species available from international culture collections. In 1936, 1937, and 1938, CSC filed US patents under the name of *Clostridium saccharo-butyl-acetonicum-liquefaciens* for two variants of this species designated gamma and delta. As part of the joint LanzaTech/JGI project [30], the genomes of 57 of the NCP production strains were sequenced. These genomes are closely related, but phylogenetic analysis used was not precise enough to determine exact relationships of these genomes with certainty. However, at least two subgroups of these strains can be identified. 

The historical records relating to the propagation of the NCP production strains do not show a good correlation with the phylogenetic relationships of the strains [28,32]. NCP were unaware that they were using two different species for solvent production, and many of the surviving spore stocks consisted of mixtures of *C. saccharobutylicum* and *C. beijerinckii*. Although these production strains have been sub-cultured numerous times since they were first isolated in the 1930s, the genomes are all very similar, indicating that the genomes are remarkably stable and highly conserved. The *C. butyricum* species constitutes a major group of butyric acid clostridia and can be distinguished from *C. beijerinckii* by major differences in their cell wall composition and their membrane phospholipids [33,34].

### 7.3. Comparative Phylogenomics of C. beijerinckii Species

There are now around 220 genome sequences for *C. beijerinckii* in GenBank, making this species of solvent-producing clostridia by far the most diverse group with the largest number of strains. Of the 270 genomes sequenced from the DJ collection, 72% were *C. beijerinckii*. The initial molecular genetic studies based on 16S rRNA gene sequencing and DNA fingerprinting indicated that at least 12 subgroups of *C. beijerinckii* could be identified [30]. The comparative genomic analysis of solventogenic clostridia published in 2017 included the genome sequences for 18 *C. beijerinckii*, and the phylogenetic analysis indicated that four clusters or subgroups could be identified within the species. This study confirmed that strains that were previously classified as *C. dolis* fall within the species and that the *C. dolis* species name is invalid [28]. The type species for *C. beijerinckii* is DSM 791, while the type species for *C. dolis* was DSM 15410. This study also confirmed that *C. pasteurianum* NRRL B-598 had been misclassified. 

The joint LanzaTech JGI project added another 194 genome sequences for *C. beijerinckii*, consisting of 160 genomes from the NCP strain collection and 34 genomes for strains originating from the international culture collection [30]. The phylogenetic analysis derived from the expanded number of genomes confirmed that the *C. beijerinckii* species consists of four distinct subgroups.

In 2021, Seldar et al. [35] published a report on the diversity and evolution of the *C. beijerinckii* species. They sequenced and assembled the complete genome of the Type strain DSM 791T and used 2308 genes of the core genome to reconstruct the phylogeny of the species. A modified representation of their phylogenetic tree showing the four subgroups is illustrated in Figure 4. Of the 246 genome assemblies of various strains available at the time, they found that strains WB53 and HUN142 were mis-identified and did not belong to the *C. beijerinckii* species. They used 237 high-quality genomes to define the pangenome of the species. The accessory genomes contained a large percentage of genes with unknown function. The use of the core genome to perform the comparative analysis and identification gave a more detailed phylogenetic tree that confirmed the separation into four groups and identified a number of sub-clusters within the groups.

The four subgroups are sufficiently demarcated and distinct to warrant at least subspecies status. The Group 1 strains form the most phylogenetically distinct cluster and contain many of the strains that are recognized as iso-propanol producers, including strains that were used in industrial production in Asia. Group 2 consists of many of the recognized industrial strains and includes the closely related A-8, A-14, and A-16 strains isolated by McCoy in the US. This cluster also exhibits genomes that are remarkably stable and highly conserved. The other two members of this group are distinct and were isolated as industrial strains in Europe. The Group 3 strains are a quite diverse assemblage and include the Type strain of *C. beijerinckii*. Most of these strains were initially isolated from sources related to food spoilage or other public health problems and are generally weak solvent producers. Although the largest number of genome sequences is from strains belonging to Group 4, except for one strain, they all originated from the NCP collection of industrial strains and are derived from a single source. The original isolates were developed by CSC in the US as their final generation of industrial solventogenic production strains. Many of their BAS subcultures were supplied to NCP during 1944 and 1945, and the descendants of these lines were propagated and used as NCP production strains. All the genomes of the Group 4 strains are very similar, and phylogenetic analysis shows that the bulk of the strains could be derived from a single ancestral isolate. Several outlying genomes and clusters can be identified, but analysis is not precise enough to determine exact relationships with certainty. In addition, the historic information relating to the propagation of the NCP production strains does not show a good correlation with the phylogenetic analysis [27,32]. The bulk of the NCP genomes are around 6.14 kb in size with a GC ratio of around 29.9, but there also appears to be a cluster with a genome size of around 6.18 kb and a GC ratio of 30 and another cluster with a genome size of around 6.01 and a GC ratio of around 29.6. Although many of the strains were sub-cultured numerous times since they were first isolated in the 1930s, the genomes are all very similar, indicating that the genomes are remarkably stable and highly conserved. This feature is emphasized by the analysis of the genomes of the seven strains in the DJ collection that were isolated in Japan in the 1950s donated by Professor Hongo. These genomes appear to be indistinguishable from the genomes derived from the CSC isolates.

### 7.4. The Pangenome of C. beijerinckii

In 2021, two groups published reports identifying and characterizing the pangenomes for *C. beijerinckii* species. In the study undertaken by Seldar et al. [35], the genome of the Type strain C. *beijerinckii* DSM 791T was sequenced and used for comparison with 237 other genomes to evaluate the diversity and evolution of the whole species. Reference genes from the Type strain were searched for sequence similarity. The genomes were used to elucidate the core genome of 2308 genes that were present in all *C. beijerinckii* genomes. An additional 12,202 genes were found in at least two genomes that formed the accessory genome. Together, they identified the pangenome containing 14,510 genes, with 5929 genes being unique and found in only one genome. The number of unique genes for particular strains ranged from zero to 516, and the number of exclusively missing genes ranged from zero to 86. Functional annotation of genes showed a different composition of the core and accessory genomes, with the core genome containing a larger proportion of genes connected to metabolism, energy production, secondary metabolites, biosynthesis, transport, and catabolism. The accessory genome had a higher relative abundance of unique genes, including genes connected to repair and defense mechanisms. While only 6.28% of genes in the core genome were not assigned any function, the abundance of unique genes in the accessory genome without known function exceeded 35%. A phylogenetic tree of *C. beijerinckii* strains was constructed using 2308 genes of the core genome, and closely related strains were collapsed into clusters (Figure 4). 

In the other investigation undertaken by Zou et al. [36], the genomes of five newly isolated *C. beijerinckii* strains from pit mud of strong-flavor baijiu ecosystems were sequenced and added to 233 genomes of other *C. beijerinckii* strains and used for pangenome analysis. The average genome size of the genomes was 6.11 Mb, and the average number of proteins was 5182. The genes in the genomes were grouped into 12,572 gene clusters. Among these, 1567 gene clusters were shared among all 233 genomes and made up the core genome. The core genome genes account for 30.2% of the average number of genes, and 8851 gene clusters were present in two or more, but not all, of the genomes. The average number of accessory gene clusters for each genome was 3153, representing 60.8% of the average number of genes for each genome. A total of 2154 strain-specific gene clusters were found in only one genome, and 179 strains had no strain-specific gene clusters. A total of 41 strains had between 1 and 10 strain-specific genes, and 13 strains had more than 10 strain-specific genes.

The genes predicted to be in the pangenome were assigned to Clusters of Orthologous Gene categories (COG), and 91.1% of the total number of genes in the core genome, 67.6% of the accessory genome, and 61.7% of the strain-specific genes were assigned COG annotations. The largest category was genes with unknown function, representing 19.4% of the core genome, 24.8% of the accessory genome, and 20.8% of strain-specific genes. Most genes in the core genome were assigned to carbohydrate transport, amino acid transport, metabolism, energy production, conversion, and transcription. The COG annotation of accessory genome and strain-specific genes revealed that genes in the largest category were involved in replication, recombination, and repair. A total of 584 genes in the accessory genome were assigned to this category. They constructed a phylogenetic tree for the 233 *C. beijerinckii* strains based on concatenated core genome alignments using a maximum likelihood algorithm. In this tree, the five newly isolated strains were located in a separate subbranch of their own.

The two studies showed some differences. The pangenome reported by Seldar et al. [35] consisted of 14,510 genes, with a core genome of 2308 genes and an accessory genome of 12,202 genes, with 5929 of the genes being strain-specific. In the study by Zou et al. [36], the pangenomes of genes were grouped into 12,572 gene clusters, with 1567 gene clusters in the core genome and 3153 gene clusters in the accessory genome, with 2154 gene clusters being strain-specific. The phylogenetic tree constructed by Zou et al. also identified the same four Groups of *C. beijerinckii* but placed Groups 1, 2, and 3 strains together as separate from Group 4 strains. In addition, a significant number of the strains listed in the tree appear to have been mis-labelled or mis-coded. They also identified an additional group of sub-species of *C. beijerinckii* comprised of recently isolated strains in China.

### 7.5. The Future Potential of Comparative Genomics and Phylogenetic Analysis 

The availability of over 400 genomes for the solventogenic and closely related clostridial species has opened the way for large-scale comparative genomic and phylogenetic analysis that has the potential to reveal hitherto unknown and unreported phenotypic features and processes that could be utilized and applicable in biotechnology.

## 8. Comparative Phylogenomic Analysis of *C. beijerinckii* Prophages

As an example of the use of comparative phylogenomic analysis, a survey was undertaken for this review on the occurrence and potential role of prophages in the solvent-producing clostridia. With over 400 fully sequenced genomes available in public databases, it has been possible to undertake a general assessment and analysis of prophages in the 10 species that were investigated.

Phages are abundant in nature, and their number globally is estimated to be approximately 10^31^ particles, outnumbering their host. Tailed double-stranded DNA phages belonging to the *Caudoviricetes* constitute ~96% of all the known DNA phages. The class *Caudoviricetes* includes phages with three major morphologies: siphoviruses with long non-contractile tails, myoviruses with long contractile tails, and podoviruses with short non-contractile tails. Phages can either be virulent or temperate, the former performing lytic infections while the latter are able to perform both lytic and lysogenic infections. During a lysogenic infection, the phage genome is often inserted into the bacterial host’s chromosome as a prophage at a specific attachment site typically recognized via a phage-encoded integrase. In the quiescent state, the prophage genome is replicated alongside the bacterial genome during cell division. The prophage has a repressor that prevents phage excision under normal conditions but is reversible. If the cell undergoes stress, such as UV radiation damage, the repressor activity is reversed, and the phage resumes the lytic pathway of replication. To force phage excision, chemical stressors such as mitomycin C can also be used. The reaction to a signal triggering the SOS response is thought to be responsible for the activation of the prophage.

The identification of prophage genomes in bacterial host genomes has been greatly enhanced by the development of tools such as PHAST and PHASTER. Identification can be challenging due to the presence of decaying prophage remnants that have lost many of the conserved phage components, such as structural genes, although these remnants could still influence their host cells even though they cannot replicate or produce complete infectious particles.

Very little is known about the distribution, diversity, and extent of prophages in clostridial species. The one exception is *C difficile.* As an important opportunistic pathogen, the occurrence of prophages in this species has probably been the most extensively studied [37,38,39,40]. Over 1300 *C. difficile* genomes have been fully sequenced and are available in public repositories. Thousands of additional genomes have also been sequenced and have been made available through collaborative research. A large number of virulent and lysogenic phages infecting *C. difficile* have been described over the last decade, and many complete phage genomes are available in public databases. Most of the prophages are members of the myoviruses, while siphovirus prophages are far less common. Large prophage genomes maintained as stable extrachromosomal DNA have also been recorded.

### 8.1. The Physiological Roles of Prophages

Most of the well-known prophages that have been characterized can be induced in the laboratory using agents known to damage the DNA that is mediated by the SOS global regulatory network. During prophage replication, the transcriptional program of the host is profoundly restructured, and metabolic resources are redirected toward phage replication. During the lysogenic cycle, prophage genomes are generally quiescent, and minimal gene transcription is observed from the prophage itself [37]. Only a few gene products are required to establish and maintain lysogeny.

The induction of prophages and phage-related elements has also been identified as having other important physiological roles. The *skinCd* is a putative prophage remnant similar to *skinB* sequences identified in *B. subtilis* that interrupts the coding sequence of *SigK*, a sporulation-associated alternative sigma factor. The excision of the *skinCd* has been shown to be important during the sporulation process.

Quorum sensing is used to coordinate specific phenotypes at the level of the whole population in assessing cell density. The quorum sensing (QS) system of *Staphylococcus aureus* is the most well-characterized QS system in Gram-positive bacteria, and it regulates the expression of hundreds of genes, including exotoxins and surface proteins. At least two types of QS systems have been described in *C. difficile* and several other bacterial pathogens and have been implicated in virulence by coordinating toxin secretion, biofilm production, motility, and sporulation. It is possible that QS signals detected by the phage-encoded genes could lead to prophage induction. This suggests that the prophage could employ QS to determine the most favorable time to initiate a replication cycle to ensure its successful propagation into the bacterial population.

Some prophage genomes have been shown to carry cargo genes unrelated to the phage replication cycle, and their expression is often independent of the phage circuitry and occurs during lysogeny. These genes often code for virulence factors, including toxins, superantigens, and hydrolytic enzymes. There is some evidence that genes encoded by some prophages can influence sugar uptake and energy metabolism with a possible impact on growth kinetics.

### 8.2. The Diversity and Distribution of Prophages in Solvent-Producing Clostridia

The majority of species and strains of solvent-producing clostridia investigated were found to encode 2–4 prophages, but some strains encoded 5–7 prophages. Most of the prophages are classified as myoviruses, and many exhibit the typical myovirus R-type architectural structure consisting of a structural cassette containing an integrase followed by a capsid gene module, a tail gene module, and a lysis gene module. The prophage upstream region is inserted into the host genome by either a Resolvase or integrase gene, followed by a regulatory gene module that can be variable. Every strain included in the survey harbored at least one prophage of this type, with some strains harboring 2–4 prophages that encode an R-type structural cassette. In the strains of *C. beijerinckii* that have been investigated, the predominant 40,000 bp R-type prophage was observed to encode for the production of tailocin particles. (see below “8. Comparative phylogenomic analysis of *C. beijerinckii* tailocins”). This has not been determined for the other species. The domesticated prophage genomes that are also present in all the strains are much more variable, and as yet, it has not been possible to identify any prophage that might be classified as a siphovirus.

The most informative perspective regarding the nature of these resident prophages can be gained by looking at the prophages encoded by closely related strains. Examples of these are the 160 DJ genome sequences in public databases for NCP Group 4 *C. beijerinckii* industrial strains. There are also a dozen or so genomes for the related McCoy Group 2 *C*. *beijerinckii* strains. The 57 DJ genome sequences for the NCP *C. saccharobutylicum* industrial strains and 12 genome sequences for *C. acetobutylicum* strains also provided valuable information.

All of the NCP Group 4 *C. beijerinckii* genomes, including those isolated independently in Japan, encode for an identical R-type tailocin prophage that has been rigorously maintained in every strain. Every Group 4 strain also encodes a second unrelated 38,993 bp prophage. In addition, there were two other prophages harbored by these strains. One of these occurred as either a full 58,548 kb genome or as a truncated 41,491 bp genome. The fourth prophage also occurred in three different genome sizes of 49,046 bp, 27,636 bp, and 14,253 bp. As well as the standard complement of four resident domesticated prophages, one strain encoded an additional acquired prophage genome.

A similar pattern emerged for the related McCoy Group 2 *C. beijerinckii* strains. These genomes encoded four different prophages—three of which contained the typical R-type structural framework. In addition, three strains belonging to the former *C. madisonii* cluster encoded for an additional three or four prophages probably acquired during phage infections that occurred in the 1940s at the plant in Puerto Rico.

The 57 DJ genome sequences for the *C. saccharobutylicum* industrial strains also exhibited a similar pattern but with some differences. Every strain contained a tailocin with the typical R-type structural framework, but these occurred in five different genome lengths ranging from 52,015 bp down to 30,399 bp. Four strains also encoded a second tailocin-like genome of 38,384 bp. These DJ genomes also harbored two other prophage genomes that also exhibited different lengths. The greater variation in the prophage profiles found in the *C. saccharobutylicum* species is possibly related to the two variants isolated and patented by CSC.

The genomes of 15 strains of *C. acetobutylicum* were also investigated. With the exception of the GXAS188_1 strain isolated in China that had a completely different profile, the other 14 strains all exhibited similar prophage profiles. Each strain is encoded for a prophage genome of around 55,000 bp that contains a typical R-type structural framework cassette. In addition, all 14 strains harbored a smaller unrelated 25,258 bp prophage that contains lysis genes. A surprising and remarkable observation is that although these strains were isolated in Europe, Britain, North America, South America, Japan, and China at different times, and many have been sub-cultured thousands of times, the prophage genome profiles were similar. This indicates that these prophage genomes are remarkably stable, highly conserved, and likely to be of ancient origin. The distribution pattern and relationships of the key prophage genomes are illustrated in the heatmap in Figure 5.

### 8.3. Extrachromosomal Prophage Elements Encoding Jumbo Phages and Giant Particles

Two extrachromosomal prophage elements that encode for a jumbo phage and a large inducible tail-like structure were also identified. The CMX jumbo phage was originally isolated from a phage infection in an Industrial fermentation in Puerto Rico [41]. The plant utilized *C. madisonii*, licensed by McCoy. The CMX temperate bacteriophage was recovered from the lysogenic *C. beijerinckii* 4J9 strain (formerly *C. madisonii*) following mitomycin C induction and was shown to undergo both lytic and lysogenic cycles [42,43]. CMX is a siphovirus and has a large isometric head with 100–110 nm diameter and a long noncontractile tail. It was shown to be flagella-trophic with primary adsorption to the flagella of the sensitive bacteria (Figure 6A). The CMX prophage is not integrated into the host chromosome but is maintained as circular extrachromosomal DNA with a genome size of 229,124 bp with around 280 ORFs. The low copy number and stability of the prophage indicate that in the lysogenic state, the CMX genome is actively partitioned between daughter cells. The presence of the lysogenic prophage was shown to affect host cell morphology, metabolism, and solvent production. After induction CMX gave a burst size of about 90–95 and had a narrow host range restricted to *C. beijerinckii* Group 2 strains [42,43]. Phages belonging to this same group appear to be quite broadly distributed and include the “giant” *C. botulinum* C-neurotoxin-converting phage, as well as examples occurring in *Bacillus* and *Staphylococcus* species [44].

In addition to producing tailocins, when induced, the *C. saccharoperbutylacetonicum* strain N1-4 also produces occasional giant tail-like particles consisting of straight, thick rods about 450 × 24 nm consisting of a core, a contractile sheath, and a basal structure with three thick, wavy, whip-like filaments of about 230 × 8 nm referred to as X particles [45,46]. The X particles occurred in both the extended and contracted forms and were often joined end to end. X particles were extremely rare in noninduced cultures (Figure 5C and Figure 6B). Similar particles resembling the tails of large phage-like particles have also been reported in *B. megaterium* that also had full or empty heads of about 90 nm in diameter and tails that were 425–450 nm long with three or four long, wavy tail fibers. The phage could not be propagated and is probably defective [47]. Structures resembling these structures have also been observed in EM preparations of the NCP P193 and P258 production strains. The *B. cereus* phage Bace-11 is one of the largest bacterial viruses known and has an isometric head and a contractile tail of about 500 nm in length with whip-like fibers, similar to giant phage tail structures observed in the other species. A viable *B. cereus* phage that was morphologically identical to the phage-like structures of *B. megaterium* has also been reported. The N1-4 strain has been reported to contain a mega-plasmid of 136,188 bp, and the X particles appear to be encoded by an extrachromosomal element that shows homology with the *Bacillus* phage genomes [28,31]. These jumbo phage-like entities have been termed “living fossils” and are suggested to represent a phage line of very ancient origin that arose before the divergence of clostridia and bacilli [47].

### 8.4. The Role and Impact of Prophages

The genomes of many prophages encode genes that are inferred to influence their host. Some prophages have been shown to have an impact on lifestyle and biology in subtle ways, depending on the environment and the genetic background of the host [37,48]. Regrettably, many of the phage genes have no homologs in databases or have no assigned function. Multiple regulatory genes have been identified in many of the prophage genomes that could potentially interfere with and influence host regulation, suggesting that their impact on the host can be complex and often subtle. Host genomes often carry multiple prophages, and phenotypes can sometimes result from the cumulative effects of more than one prophage. Crosstalk between prophages has been observed, including between their recombinases. It is likely that a similar picture will emerge for prophages in the solvent-producing clostridia. The picture that has begun to emerge is that in both Clade 1 and Clade 2 species, each strain encodes for and maintains at least two resident prophage genomes. These genomes are highly conserved and stable and can be considered to have become domesticated by the host species for the mutual benefit of both host and prophage. In all of the species investigated at least one of the resident prophages were classified as myoviruses, and many exhibit the typical R-type architectural structure. These resident domesticated prophages probably represent phage lines of very ancient origin with regulatory functions that have become seamlessly integrated into the host regulatory networks. There appears to have been an evolutionary progression from fully replicative lysogenic phages to defective phages, to defective tailocins, and finally to lytic suicide vectors. These resident prophages appear to provide attack systems against related species, defense systems against phage infections that favor both the host cell and prophage, as well as prophage-encoded altruistic lytic suicide systems that can be induced in populations under stress so some cells can be sacrificed to enhance group survival.

In a study of the domestication of defective prophages by host bacteria undertaken by Bobay et al. [48], comparative genomics was used to study the evolution of prophages within 300 enterobacterial genomes. They concluded that bacteria frequently domesticate their prophages and that bacterial hosts select for phage-associated functions, and the integrated prophages are major contributors to the diversity of bacterial gene repertoires. Domestication of prophage components indicates that they can endow the bacterial host with molecular systems involved in secretion, defense, warfare, and gene transfer. They noted that the patterns of selection were observed not only in accessory regions but also in core phage genes, such as those encoding structural and lysis components. Several of the conserved prophages had gene repertoires compatible with the described functions of adaptive prophage-derived elements such as bacteriocins, killer particles, gene transfer agents, or satellite prophages. The size distribution of prophage elements was bimodal, suggestive of rapid prophage inactivation followed by much slower genetic modification and degradation. They concluded that many of the prophage elements were very old and probably pre-dated the split between species and clades [48].

The survey of prophages in the solvent-producing clostridia indicates that domesticated prophages most likely contributed significantly to the shaping of the host species. The evidence suggests that many of these domesticated prophages have very ancient origins and probably pre-dated the split between species and clades. Studying the interplay between the prophage genomes and the cell microbiome can have practical implications with respect to their potential to influence the outcomes of industrial fermentation processes. Clearly, there is a lot to discover about the prophages in the clostridia, and recently developed molecular tools and the availability of high-quality bacterial genome sequences will certainly foster research in this area.

## 9. Comparative Phylogenomic Analysis of *C. beijerinckii* Tailocins

A second example of the use of comparative phylogenomic analysis is a survey undertaken for this review to investigate the occurrence and role of tailocins in the solvent-producing clostridia.

### 9.1. General Features of Tailocins

Tailocins were first discovered in the 1950s and have interchangeably been referred to as “phage tail-like bacteriocins” or PTLBs. Tailocins are widespread among the eubacteria. These phage tail-like bacteriocins are high-molecular-weight bactericidal protein particles that resemble and are evolutionarily related to the tail structures of various bacteriophages without being able to replicate or produce a phage capsid. They are produced in response to SOS induction, and the tail then acts as a molecular syringe to puncture the cell envelope and kill the target cell. In general, they are only composed of proteins, and no evidence of DNA has been reported [49,50,51,52].

Based on the observations that tailocins can be produced under situations when lysogenic phages are often induced by SOS induction and released by lysis, it has been assumed that tailocins have originated from prophages that have lost the ability to replicate DNA. Further adaptations would have made them more efficient at killing cells rather than transducing DNA. This is supported by the evidence that the tail and bacteriocin functions have evolved in parallel multiple times and comprise different types. Some tailocins have diverged considerably from any known phage, suggesting that they are probably quite ancient [52].

Two types of tailocins have been identified: the R-type tailocins, which are rigid and contractile and are related to myovirus phage tail structures, and the F-type tailocins, which are flexible and noncontractile and are related to siphovirus phage tails [49,50,52].

The majority of tailocins have been isolated and characterized from various Gram-negative bacterial species such as *Pseudomonas, Listeria, Vibrio, Rhizobium, Yersinia, Serratia,* and *Proteus.* These include both R-type and F-type tailocins. The most intensively studied are the tailocins produced by *Pseudomonas* species [49].

The R-type “diffocins” from *C. difficile* constitute the only Gram-positive R-type tailocins that are widely recognized and well documented. One characteristic of the prophage genomes that encode tailocins in Gram-negative bacteria is that the phage capsid genes have been lost. This appears to be emerging as a major difference as the capsid genes appear to be highly conserved in Gram-positive bacteria [38].

### 9.2. The Biological Role of Tailocins

Integrated prophages are major contributors to the diversity of bacterial gene repertoires. The retention of resident tailocin genomes illustrates the capacity of bacteria to accommodate exogenous genetic elements and domesticate them for their own benefit. Prophages that produce tailocins are not just prophage remnants that no longer possess the reproductive capability. Rather, they can be regarded as defective prophages that have been retained and domesticated as specialized phage defense systems as well as providing inducible attack and cell lytic capability that provide a strong selective advantage for their retention. The tail structures serve as precision killer particles for interbacterial warfare. Evidence of their co-evolution suggests that the conserved domesticated prophages have had an impact on both genome evolution and biology of their host as well as other phages [49,50,52].

### 9.3. The R-Type Tailocins

R-type tailocins are produced and accumulate within the bacterial cell under SOS response conditions and are only released following cell lysis. The tailocins act as molecular puncture devices that penetrate the cell envelope of related strains and kill the attacked bacterium.

The R-type tailocins consist of a core consisting of a tube of a polymer of a single polypeptide. Attached to the core is a pointed trimeric tail spike protein that has an iron moiety at the tip. At one end is a complex baseplate structure consisting of several separate polypeptides. The core is surrounded by a sheath that also consists of a polymer of a single polypeptide. The sheath and core assembly in the uncontracted form is usually around 120 nm in length. Attached to the baseplate are tail fibers, which are composed of a single polypeptide and serve as the receptor-binding proteins. These proteins are responsible for recognition and binding to specific receptors on the surface of the target cell and are responsible for narrow target receptor specificities [53,54,55]. The similarity between myovirus phage particles and R-type tailocins is illustrated schematically in Figure 7.

The mechanism of killing R-type tailocins is related to the mechanism used by myovirus phages to translocate DNA into the cell. With tailocins, instead of DNA entering the cell, a pore is created. It has been shown that contact with a single tailocin particle is sufficient to kill a bacterial cell [51,52,53]. Once bound, binding triggers extensive conformational changes in the baseplate, accompanied by contraction of the sheath, driving the tail spike into the cell envelope and forcing the rigid inner core through the cell membrane. This creates a channel through which protons and other small ions flow, disrupting membrane permeability and causing a dissipation of membrane potential [53,54,55]. 

The target bacteria are typically quite restricted, so tailocins do not seem to serve as general antibacterial weapons. Cells of the same or related strains harboring the tailocin genomes are provided with a resistance mechanism, allowing sister cells to gain a subtle biological advantage. The cell producing the tailocin must sacrifice itself because tailocins can only be released into the medium by cell lysis. This can be viewed as a form of altruism, with cells producing the tailocin providing sister cells with a competitive advantage. Given the high potency of the tailocin, it may be that only a small fraction of a population is needed to lyse and release particles to give the surviving population a competitive growth advantage. The selective advantage that they provide to cells likely involves subtle interactions with closely related bacteria competing in tight communities. There is some evidence to indicate that some prophage genomes have lost the ability to produce tailocins but still retain a functional mechanism for induced lysis. There could still be a selective advantage for the retention of the genome by the host to maintain an inducible altruistic suicide capability [49,50,52].

### 9.4. The R-Type Tailocins Produced by Gram-Positive Bacteria

Tailocins that have been most generally recognized and extensively studied in Gram-positive bacteria are the R-type diffocins produced by *C. difficile* [38]. A particulate bacteriocin consisting of a tail-like structure with a contracted sheath designated Boticin P, produced by a non-toxigenic type E strain of *C. botulinum* has also been reported [56,57]. The *C. botulinum* type E strains are non-proteolytic and are phylogenetically related to the Clade 2 solventogenic clostridia. Of the clostridial species tested, only *C. botulinum* type E and, to a lesser extent, *C. perfringens* and *C. acetobutylicum*, but not *C. botulinum* types A, B, or F, were sensitive to the tailocin. Of 18 the type E strains tested, Boticin P was active against 13 strains. Of the 23 type E strains induced with mitomycin-C, many produced phages with icosahedral capsids and generally contracted tails as well as sheathed tail-like structures resembling typical tailocins. Induced lysis of *C. sordellii* and *C. tetani* has also been reported [51]. A 1981 survey of 34 strains belonging to 16 species of *Clostridium* described the production of a wide variety of phage or phage-like particles [58]. These ranged from intact phage virions, R-type tailocins, rod-shaped structures, sheaths, filaments, and fibrils. Some of the particles were only produced after induction with mutagenic agents, while other strains exhibited spontaneous production. Of the 34 strains tested, 25 were found to exhibit some bacteriocin-like activity. This suggests that the production of R-type tailocins could be quite common and widespread in *Clostridium* species.

### 9.5. The Production of R-Type Tailocins in Solvent-Producing Clostridia

Tailocin production in the solvent-producing clostridia was first investigated by Hongo and Ogata and their colleagues in Japan in the 1960s/1970s [59]. Their discoveries published in Japanese journals have been largely overlooked. A summary of their findings was published in a review of bacteriophages in *Clostridium* published in 1979. They identified and studied inducible defective phage tail-like particles produced by *C. saccharoperbutylacetonicum* named Clostocin O and Clostocin A. No heads were produced, and the tails occurred in both the uncontracted and contracted state after liberation by lysis of the host cell. Clostocin O only killed Clostocin-M-producing cells, and Clostocin M only killed Clostocin O producers, and the killing activity showed single-hit kinetics [59].

In the later 1990s, studies conducted at the University of Otago identified and studied the occurrence of similar tailocin particles produced by several *C. beijerinckii* strains [43]. When induced, these strains produced large numbers of tailocin particles. In most cases, the tailocin particles were present in the contracted form due to the method of preparation (Figure 8A). In addition, some strains placed in other *C. beijerinckii* groups also produced occasional empty phage heads and, more rarely, defective ghost phage particles (Figure 8B). In the induced strains, intact prophage DNA was also released into the medium from the lysed cells. Later, the sequence of the released prophage genome DNA harvested from the NCIMB 8052 strain was mapped back to a prophage sequence in the host genome. When this DNA was used as a probe, it revealed homology with most of the other tailocin producer strains investigated. The tailocins that were produced exhibited little or no bacteriocin activity against the producer strains but showed strong bacteriocin activity against one *C. beijerinckii* non-producer strain.

### 9.6. The Survey of R-Type Tailocins in C. beijerinckii 

The availability of over 300 genome sequences for solvent-producing and related clostridial strains enabled a survey of the R-type tailocins. Initially, the survey focused on the 194 *C. beijerinckii* DJ genome sequences in the NCBI database. These consisted of 160 Group 4 DJ strains from the NCP collection that included the 8 strains from Japan, as well as the 34 DJ strains belonging to Groups 1, 2, and 3, obtained from international culture collections. Additional GenBank genome sequences were also added to the survey. All the *C. beijerinckii* genomes investigated were found to contain at least one prophage that encoded sequences, such as the NCIMB 8052 genome sequence known to encode for an R-type tailocin.

### 9.7. Common Features of the Standard R-Type Tailocin Genomes in C. beijerinckii

The common standard R-type tailocin genomes ranged from around 38,150 to 41,500 bp, with one of 45,700 bp. Some of the size differences appear to be due to the methodology used to identify prophage genomes, as prophage and tailocin boundaries can be challenging to map precisely. Hence, some genomes appeared to have additional upstream or downstream genes included or omitted in the predicted tailocin.

Two slightly different configurations of the R-type genome were identified. The one type was integrated into the host genome by Resolvase 146. The other type was integrated, employing two different variants of a different integrase gene. The Resolvase 146 gene was the most highly conserved of all the prophage genes and was present in all four of the *C. beijerinckii* groups, with 15 different genome variants being identified. An example of the typical common 40,000 bp genome organization for the “Resolvase” R-type tailocin present in *C. beijerinckii* Group 1–4 strains is shown in Figure 9, illustrating the upstream regulatory region followed by the four conserved modules making up the typical R-type structural cassette.

The Resolvase 146 gene was also present in the *C. saccharoperbutylacetonicum* N1-4 strain tailocin, as well as the *C. butyricum*, strains investigated. The tailocin genomes headed by the other type of integrase were only present in *C. beijerinckii* Group 1 strains, where nine prophage variants were identified. Several Group 1 genomes contained two tailocin genomes, with one belonging to the Resolvase 146 type and the other belonging to the integrase type. One strain contained two different Resolvase 146-type tailocin genomes. The two forms all encoded very similar structural gene cassettes, with the main differences being observed in the upstream regulatory genes.

Although the prophage genomes, in many cases, showed very little overall DNA similarity, all these genomes exhibited a high level of similarity with respect to the function and order of the genes encoded by the prophage genome. The common genome framework is headed by an integration gene followed by the distal regulatory gene module. The regulatory gene module is then followed by a conserved myovirus R-type cassette of structural genes consisting of an integration module, a head gene module, a tail gene module, and a lysis gene module. 

### 9.8. Non-Standard Larger C. beijerinckii R-Type Prophage Genomes

In addition to the common standard R-type genomes of around 40 kb, some *C. beijerinckii* strains also encoded larger prophage genomes that also contained the conserved R-type structural gene cassette. These included two genomes of around 59 kb, as well as seven prophage genomes of around 70 kb. There was also one prophage with a genome of around 104 kb and another of around 141 kb. Most of the strains that harbor larger prophage genomes also encode for a prophage of the common standard 40 kb type, but there were two strains that only appeared to encode for larger prophage genomes. Although all the larger genomes contained the conserved structural gene cassette, the position of the cassette varied widely, being located near the distal, middle, or terminal end of the prophage genome. The larger genome coded for numerous additional genes of unknown function, and it has not been possible to determine if these are additional genes that have been incorporated into these larger prophage genomes or if these are genes that extend beyond the actual prophage boundary and have been wrongly predicted as encoded by the prophage. The significance and possible biological role of these larger prophage genomes are unknown.

### 9.9. The Presence of R-Type Structural Cassette in Other Solvent-Producing Clostridia

Prophages that encode a myovirus R-type structural cassette were found to occur in every species of solvent-producing clostridia investigated. These were first identified in *C. beijerinckii* and were then found in the other saccharolytic Clade 2 *C. saccharoperbutylacetonicum, C. saccharobutylicum*, and *C. butyricum* species. Later, they were also found to be present in the Clade 1 *C. acetobutylicum, C. aurantibutyricum, C. felsineum, C. roseum,* and *C. pasteurianum* species. Prophages encoding variants of the structural cassette were also found *in C. tetanomorphum*, *C. thermocellum*, and *C. phytofermentans.*

The *C. saccharoperbutylacetonicum* N1-4 strain that produces Clostocin O tailocin particles contains a prophage genome sequence originally reported by Schueler et al. [46]. The N1-4 genome sequence exhibits very close similarity to the standard R-type Resolvase 146 tailocin genomes found in all four groups of *C. beijerinckii*. The *C. saccharoperbutylacetonicum* N1-504 sub-species was also found to encode an R-type tailocin-like prophage genome of 39,360 bp that differed from the N1-4 strain in that it belonged to the integrase type. N1-504 also encoded a second larger genome of 59,271 bp that contained an insert in the capsid module of an R-type structural cassette. A third prophage genome of 48,720 bp was unrelated. The five DJ genomes of *C. butyricum* strains that were investigated all encoded prophages with the standard R-type genome of around 40,000 kb belonging to the Resolvase 146 type in addition to harboring three or four other prophage genomes Figure 9.

The 57 strains of *C. saccharobutylicum* were also encoded for prophage genomes containing *Myoviridae* R-type structural cassettes, but these differed significantly from those occurring in the other three saccharolytic species. All the strains contained similar genomes with the typical R-type structural framework, but the genomes occurred in five different lengths ranging from 52,015 bp down to 30,399 bp. Four of the strains also encoded a second tailocin-like genome of 38,384 kb. It has not been established if this species produces functional tailocins.

The genomes of 15 strains of *C. acetobutylicum* were also investigated. With the exception of the GXAS188_1, a strain isolated in China that had a completely different profile, the other 14 strains all exhibited very similar prophage profiles. Each strain encoded for a prophage genome of around 55,000 bp that contained a typical myovirus R-type structural cassette. In all the strains, the phage genomes had Resolvase 146 genes at one or both ends. In most cases, the R-type structural framework cassette lacked some tail genes, and some genomes had extra upstream and/or downstream genes while other genomes were slightly truncated. The absence of tail genes would suggest that these strains might be unable to produce tailocin particles.

The other related Clade 1 *C. aurantibutyricum, C. felsineum,* and *C. roseum* species all contained prophage genomes of the integrase type of between 30,000 and 37,000 bp that contained a typical myovirus R-type structural cassette. Most of these genomes had reduced numbers of regulatory genes, and some were missing structural genes. One strain encoded a larger genome of 63,500 bp, headed by a Resolvase 146 gene, that also contained an R-type structural cassette. The *C. pasteurianum* strains that were investigated all encoded prophages with the standard R-type genomes of around 36,700 bp belonging to the integrase type. The *C. tetanomorphum* strains that were investigated all encoded prophages with the standard R-type genomes belonging to the Resolvase 146 type.

### 9.10. The Regulatory Gene Module in the Standard R-Type Genome

The regulatory gene modules in the standard R-type genomes in the different species and strains were found to be much more diverse compared to the genes in the structural cassettes. The number of regulatory genes ranged from 20 to 27, and in many cases, these regulatory genes had no homologs in databases and had no assigned function. There were some conserved genes, but there were a number of genes that appeared to be unique to specific strains. The regulatory genes in the Resolvase 146 type genomes and the integrase type genomes also showed distinct regulatory framework pattern differences. Of the spectrum of conserved genes that have suggested annotations, their presence in the individual strains was quite variable.

The genes coding for integration into the host genome located at the beginning of the R-type tailocin genome consisted of a resolvase, an integrase, or a site-specific recombinase. In a number of strains, the tailocin-R prophage genome was found to be inserted into a host ABC transporter gene. The regulatory genes that have a possible assigned function appear to fall into two broad categories. The one category included the genes that can be assumed to be related to DNA synthesis and repair. These include replication origins, DNA replication, DNA primase, DNA helicase, DNA polymerase, DNA segregation, DNA structure-specific endonuclease, DNA repair endonuclease, 3’ 5’ exoribonuclease, ssDNA binding and annealing, RNA polymerase, sigma factor, primosomal replication, replisome organizer, Spo0J partition family, ATPases, and methylase.

The second category of regulatory genes appeared to be related to prophage induction and repression of transcriptional control. Possible suggested functions include two component regulators, transcription regulators, DNA binding repressors and anti-repressors, RecA signal transduction, stress response, oxidative stress response, Spo0E-like, toxin/antitoxin, and growth arrest genes.

In the Resolvase 146 type genomes, there are possible early expression genes that encode a transcriptional regulator that could be involved in the SOS response or that function as a signaling module in the DNA damage response. These show some homology with the XRE family of transcriptional regulators with similarities to the repressor proteins CI of phage lambda and xenobiotic response element (XRE) of *B. subtilis*. Both are involved in lysogenic-lytic decision-making. Some of these show similarities with BRCT domains that can serve as integral signaling modules in the DNA damage response. The regulatory signaling gene appears to be linked to repressor and anti-repressor with HTH DNA binding domains that are transcribed in opposite directions. The genes could be involved in the control of lysis-lysogeny decision-making.

### 9.11. The Common Features of the Myovirus R-Type Structural Gene Cassette

The standard framework of myovirus R-type structural gene cassette appears to be highly conserved and was found in prophages of all 10 of the species investigated. The genes in the structural cassette show a remarkable conservation of gene order and function in every species. However, a heat map of the prophage genomes that encode the structural cassette showed that only a very low level of homology exists at the level of the DNA sequences. The heat map indicates a possible phylogenic relationship of 25 genomes that could be assigned to five groupings. The genomes within each group, in most cases, showed only a very low level of homology, and homology between the five groups was almost non-existent. This means that it is not possible to identify and classify these genomes based on DNA homology. This is exemplified by the observation that each gene category present in the structural cassette can have up to nine DNA variants ranging in size and varying from a high level of DNA similarity to almost no shared detectable DNA similarity.

In all the standard R-type prophages investigated, the genes in the structural cassette are all transcribed in the same direction and commence with an integration module of two to five genes headed by integrase and usually encodes an HNH homing endonuclease. This second type of integrase gene is present in all prophage genomes with the typical R-type structural cassette and appears to be involved in some way in integrating the cassette itself into the prophage genome. The genes that come first in the phage head modules consist of a small and a large terminase gene that is responsible for the recognition and encapsidation of DNA. The terminase complex is always followed in order by the genes encoding for the phage portal protein, a peptidase/prohead protease, a capsid protein, a head-tail joining protein, and a phage head closure protein.

The genes in the phage tail module occur in the order of a sheath protein, a tail tube protein, a phage tail tape measure protein, a sipho-tail protein (in some cases), a cell wall binding protein, and one or more minor prophage tail proteins ending in a baseplate protein that is followed by up to three tail fiber genes. The large tape measure protein is responsible for determining the length of the tail tube and has membrane-spanning regions that have recently been shown to insert into the cell membrane and play a role in forming a pore in the inner membrane. In a number of genomes, the structural genes are interspersed with other ORFs of unknown function, and some are thought to possibly act as assembly proteins or chaperones that help catalyze the formation of the structure.

The lysis module is always positioned downstream of the tail gene module. The genes in this module show more variability and can include genes encoding an acyl transferase, a methyltransferase, cell wall binding proteins, and large trans-membrane proteins. In virtually all genomes, the lysis module encodes for proteins that function to release the particles into the surrounding medium. Both holin and endolysin proteins are present; the holin forms a pore through the cell membranes, and the endolysin breaks down the peptidoglycan layer, resulting in cell lysis. In some genomes, in addition to an amidase or glycohydrolase, there is an additional endolysin or holin, and there may also be a lipoprotein gene.

### 9.12. Summary of the Occurnace of Tailocins in the Clostridia

The survey of the occurrence of tailocins in the solvent-producing clostridia revealed that all the strains belonging to the *C. beijerinckii* species have the potential to produce R-type tailocins. The presence of prophages that encode for an R-type structural cassette suggests that other species investigated could also have this potential. The ability to produce R-type tailocins encoded by related myovirus prophages is probably widespread amongst other species of clostridia and possibly extends to *Bacillus* species and other Gram-positive genera. It can be postulated that maintaining a flexible and dynamic tailocin attack and defense system provides a competitive advantage by suppressing the numerous closely related species and strains, all competing for the same limited resources. This, coupled with the altruistic suicide system, could make tailocins an essential component in the arsenal of these versatile, ubiquitous bacteria.

It was observed that some strains harbor two or three different but apparently functional R-type tailocin genomes. In addition to the common 40 kb tailocin genomes, a number of strains also encode for larger prophage genomes that harbor clearly recognizable R-type structural cassettes. The presence and role of these prophages is an intriguing mystery.

Another finding from the survey is that the R-type tailocin prophage genomes in the Gram-positive species differ from the Gram-negative tailocins in that they have retained the full complement of structural genes even though many strains only produce inducible tailocin particles. Some strains also produce much lower numbers of empty heads and or defective phage ghost particles. The detection of free DNA after lysis matching the complete prophage genome suggests that tailocin production could represent an evolutionary intermediary stage at which the prophage is still relatively intact and could still replicate, but the host has disrupted one or more of the critical steps either for capsid head formation or for capsid-tail junction linkage.

Another observation is that although many of the structural genes do not show many similarities at the DNA level, the full complement of the key structural genes and their exact order is maintained in virtually every tailocin-R prophage genome investigated. Specifically, there appear to be discreet variants for each gene that have been selected on a mix-and-match basis. This suggests the variants have an ancient and complex evolutionary history.

From an applied practical aspect, the production of killer particles directed against closely related strains and species could have a significant bearing on co-culture systems. The possibility of the premature induction of autolytic suicide activity at the end of exponential growth during the transition to solventogenesis could also have serious implications for an industrial process where environmental conditions become less than optimal.

## 10. Future Outlook and Prospects

The clostridia are destined to play a major role in new developments in biotechnology. It is intended that the information presented in this review will contribute to future endeavors in the field. A big advantage the clostridia provide is that they can convert various substrates, including first-generation biomass crops, second-generation lignocellulosic biomass, and third-generation C1 gases into feedstocks and other high-value products. These could include secondary metabolites such as antibiotics, lantibiotics, bacteriocins, tailocins, and probiotics. Clostridia could also play a role in the development of single-cell protein for animal/human feed ingredients and supplements. These attributes make clostridia attractive candidates for biorefinery applications as microbial cell factories for the sustainable utilization of the three generations of feedstocks for short- and medium-chain ester production. Particular attention has been given to the challenges of using agro-industrial wastes as feedstocks. However, the complexity of achieving financially viable supply chains for lignocellulosic substrates, coupled with the cost of deconstruction and processing, makes it difficult to achieve efficient production routes to utilize these raw materials. In the past, the clostridia were considered largely genetically inaccessible, but this has now changed. Considerable progress has been made with genetic tool development and metabolic engineering, and there is now an array of genetic tools available to researchers. CRISPR-Cas9 genome engineering has been successfully applied for controlled gene expression, and several new synthetic biology tools and technologies have proved useful in accelerating strain development to expand the product spectrum. Promising technologies such as metabolic and process engineering are being developed to address these various challenges. Cell-free pathway prototyping has also been used to accelerate strain engineering for increased product titers and new chemical products. The use of clostridial co-culture systems for consolidated bioprocessing includes systems involving cellulolytic and solventogenic clostridia as well as synthetic clostridial syntropy to enable CO_2_ fixation and achieve superior metabolite yields. These advances should accelerate future developments and open the way for commercial process scale-up for conversion opportunities and the rollout of new products. The rising levels of greenhouse gases in our atmosphere and oceans leading to climate change pose significant environmental, economic, and social challenges globally. The development of carbon recycling in a circular economy for the sustainable production of chemicals and biofuels from non-fossil carbon sources could become a key technology for reducing greenhouse gas emissions. The gas fermentation field has developed rapidly over the past decade. Gas fermentation using carbon-fixing microorganisms that enable the capture and conversion of greenhouse and waste gases into useful products has already been commercialized and offers an economically viable and scalable solution with unique feedstock and product flexibility.

## Figures and Tables

**Figure 1 microorganisms-11-02253-f001:**
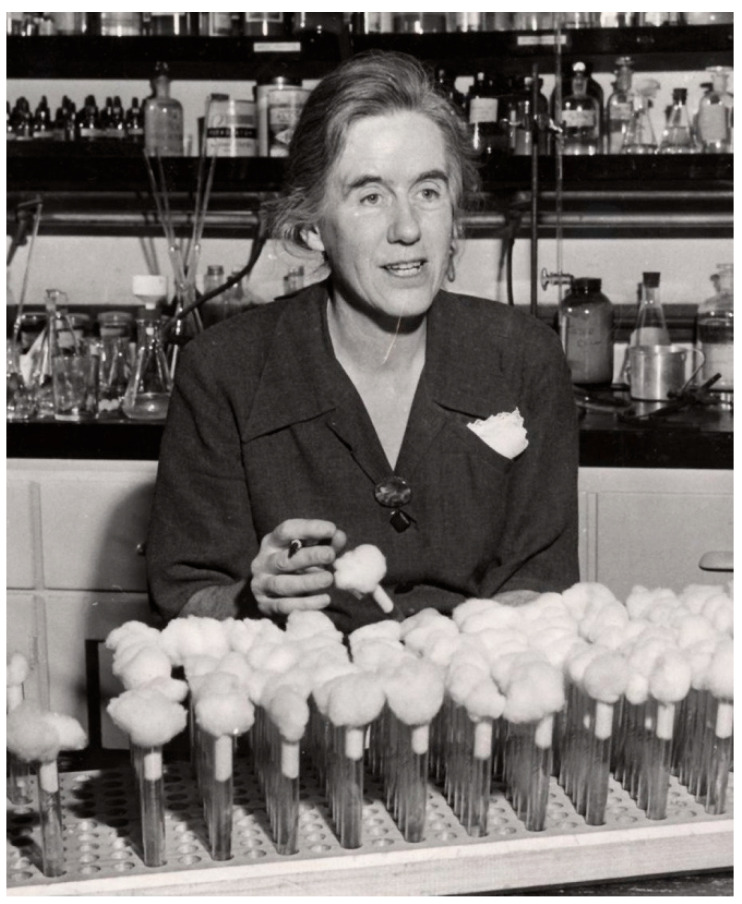
Elizabeth McCoy, pictured with her spore stocks in her bacteriology lab in 1953.

**Figure 2 microorganisms-11-02253-f002:**
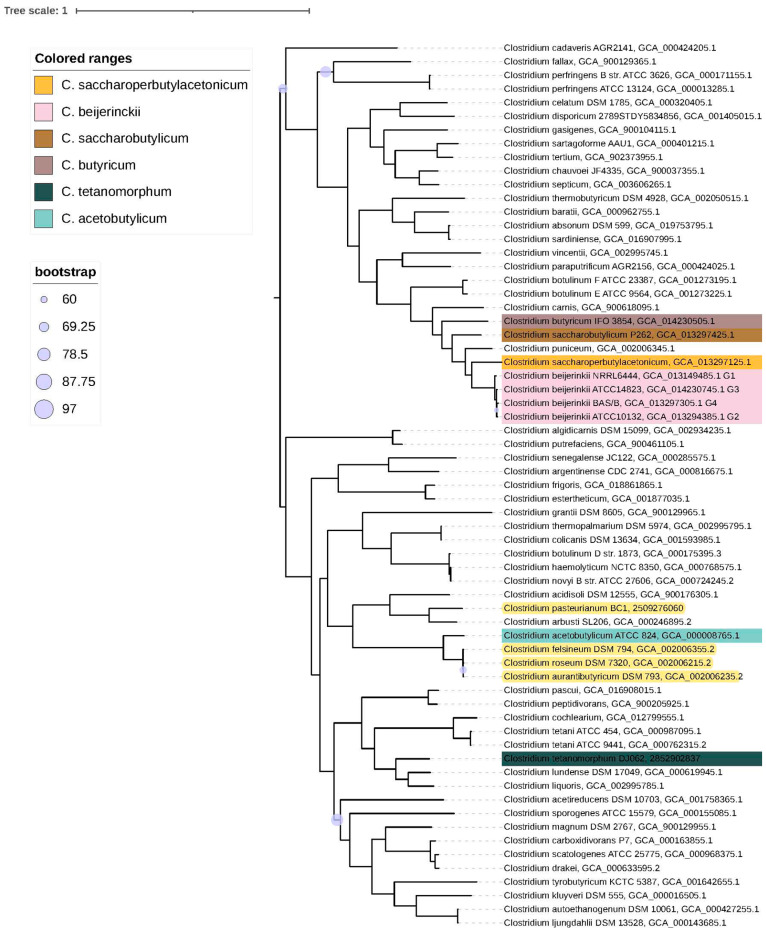
Phylogenetic tree showing the relationship of 64 different *Clostridium* species within *sensu stricto* (rRNA clostridial cluster). The solvent-producing species are highlighted in color and include an example of each of the four subgroups of the *C. beijerinckii* species. Other Clade 1 species *C. pasteurianum, C. felsineum, C. aurantibutyricum* and *C. roseum*, are highlighted in yellow. The tree was reconstructed using the neighbor-joining method based on the pairwise comparison of approximately 1340 nt. Bootstrap values (90%), expressed as a percentage of 1000 replications. Bar, 1% sequence divergence.

**Figure 3 microorganisms-11-02253-f003:**
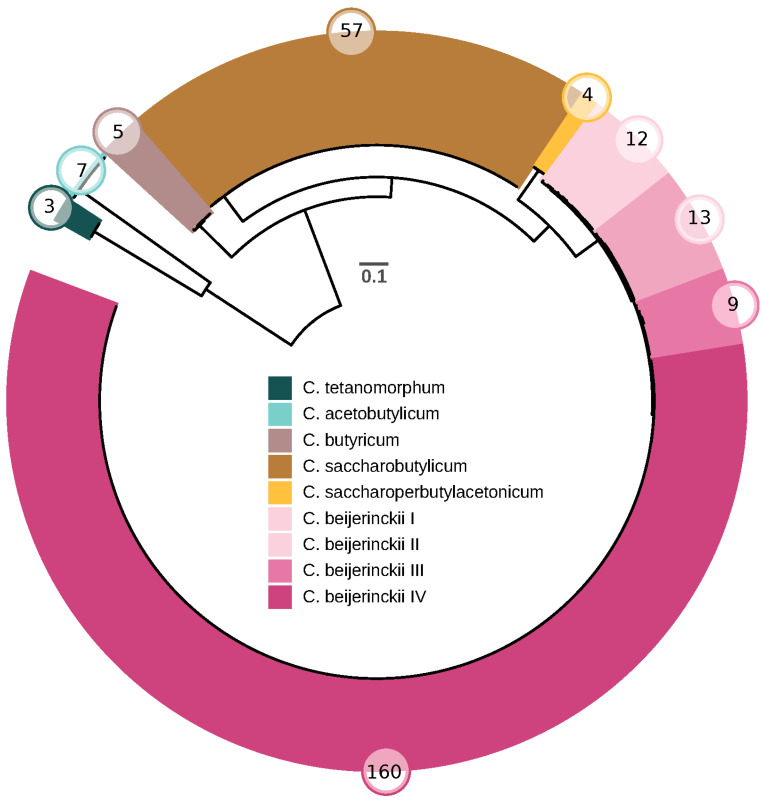
Circular phylogenetic tree showing the relationship of the 6 *Clostridium* species sourced from the DJ culture collection giving the number of strains for each species. The phylogenetic tree was constructed using a concatenated alignment of 118 single-copy panorthologs that were least affected by horizontal gene transfer (IQ-Tree, GRT+R10). Bar,1% sequence divergence.

**Figure 4 microorganisms-11-02253-f004:**
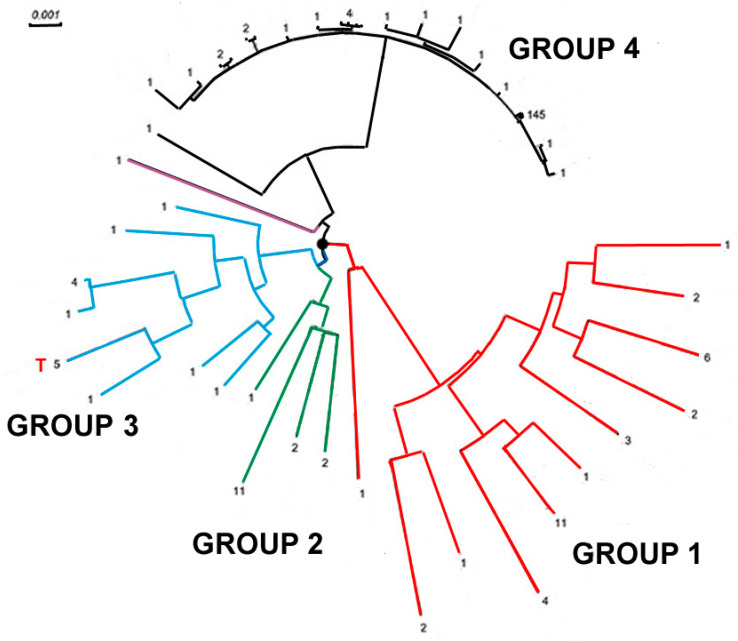
A modified phylogenetic tree of *C. beijerinckii* species reproduced with permission of Sedlar et al. [33]. The original tree has been modified to show the relationship of the four *C. beijerinckii* subgroups within the species. The branch containing the Type strain DSM 791T is marked by T. The original tree was reconstructed using 2308 genes of the core genome, present in every strain. Branches that were shorter than 1% of the maximum branch length were collapsed into clusters.

**Figure 5 microorganisms-11-02253-f005:**
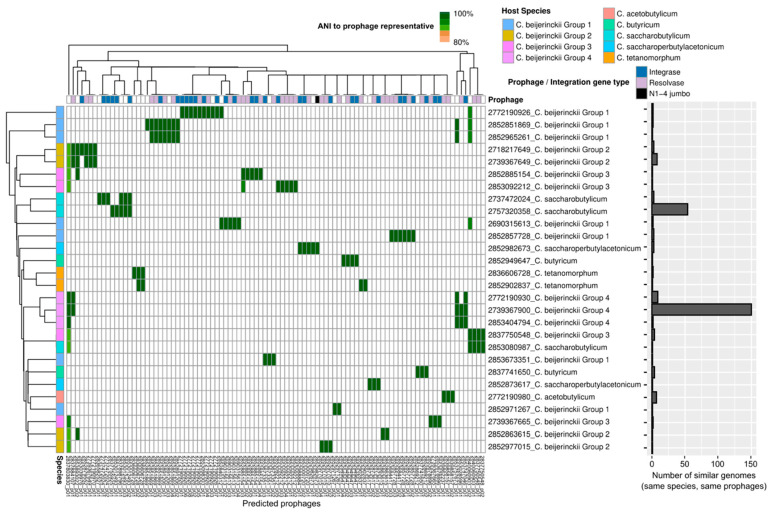
A heatmap showing the distribution of prophages across selected *Clostridium* genomes. Prophages/tailocins were predicted with VirSorter 2, manually curated to remove false-positive predictions, and clustered into a non-redundant dataset using standard cutoffs of 95% ANI and 85% AF. The distribution of representative (non-redundant) prophages across genomes was then calculated based on BlastN (minimum 80% coverage). For each clostridium species/group, genomes with the same prophage content were gathered and are represented by a single genome in the heatmap. The total number of genomes within the same group and with the same prophage content is indicated in the bar graph to the right of the heatmap. For predicted prophages, the N1-4 jumbo prophage is highlighted in black, while other prophages are colored based on the integration mechanism identified (“Resolvase”: pfam00239, “Integrase”: pfam00589/pfam14659).

**Figure 6 microorganisms-11-02253-f006:**
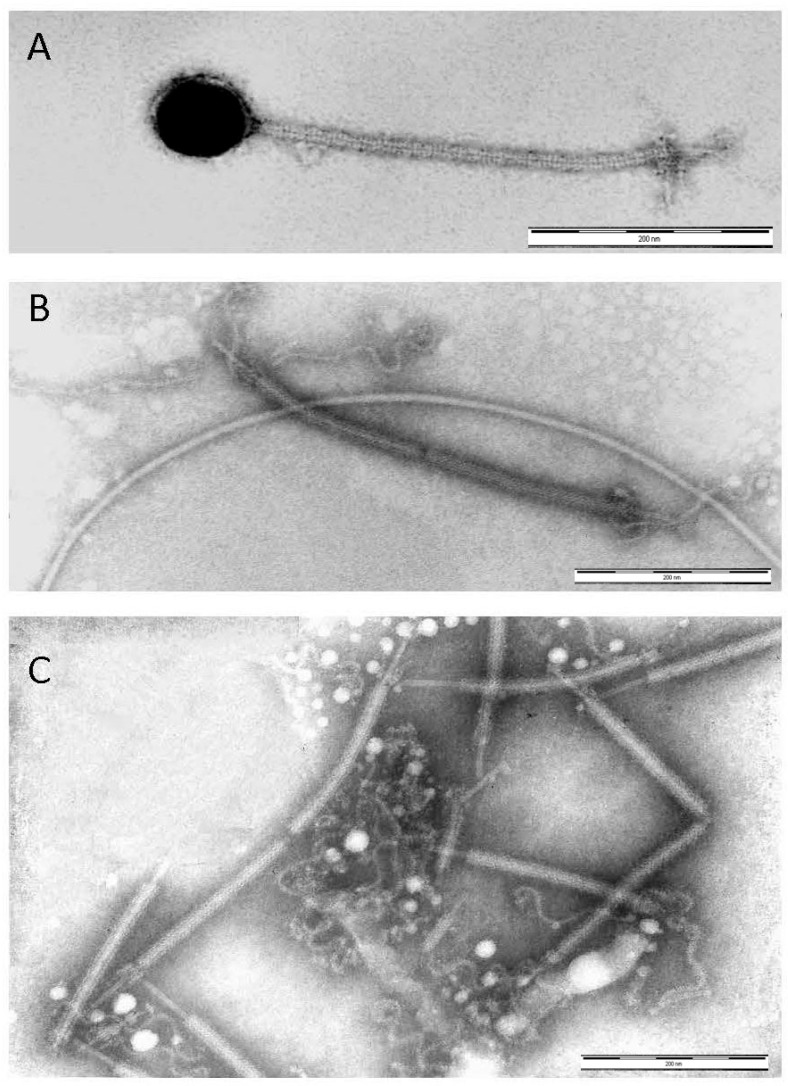
(**A**) An electron micrograph of the CMX phage induced from the *C. beijerinckii* 4J9 strain using mitomycin C. (**B**,**C**) Electron micrographs of the jumbo tail-like structures from the *C. saccharoperbutylacetonicum* N1-4 strain after induction using mitomycin C. Samples were negatively stained with 2% (*wt/vol*) aqueous uranyl acetate. Bar = 200.

**Figure 7 microorganisms-11-02253-f007:**
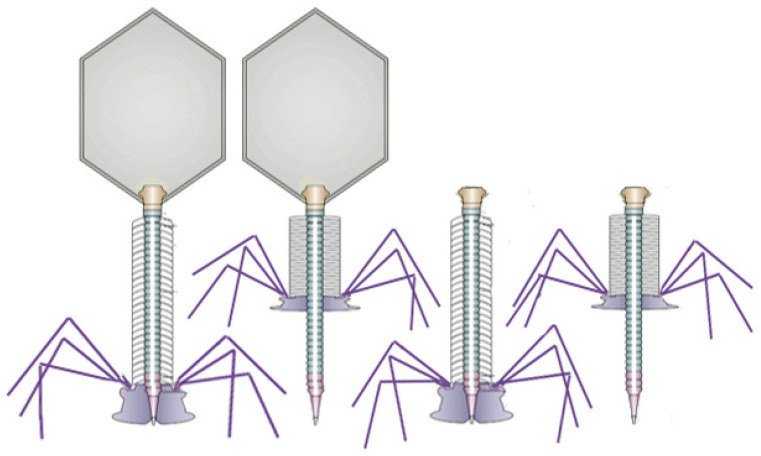
A schematic representation of a myovirus phage with a contractile tail and R-type tailocin showing the structural components consisting of an icosahedral head, neck, sheath, tail tube, baseplate, tail fibers, and tail spike.

**Figure 8 microorganisms-11-02253-f008:**
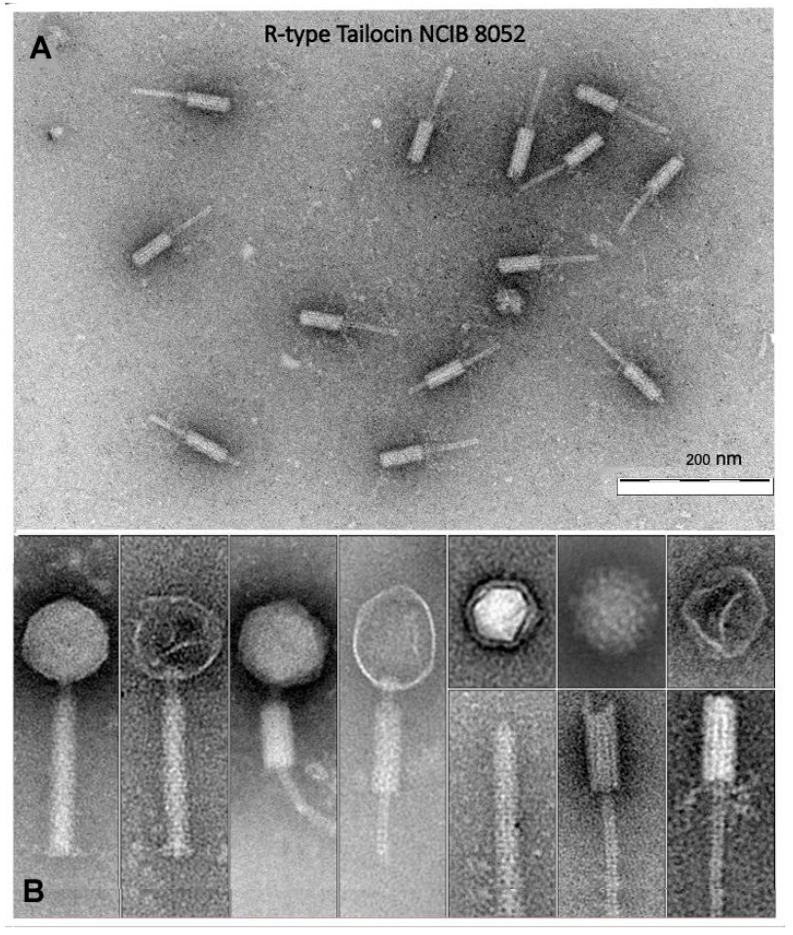
(**A**). An electron micrograph of *C. beijerinckii* NCIMB 8052 tailocins after induction with mitomycin C. (**B**) Variations in particle morphology observed in some *C. beijerinckii* Group 1–4 strains that included capsids and ghost particles. In each case the majority of the particles consisted of typical contracted tailocin particles. Samples were negatively stained with 2% (*wt/vol*) aqueous uranyl acetate. Bar = 200 nm.

**Figure 9 microorganisms-11-02253-f009:**
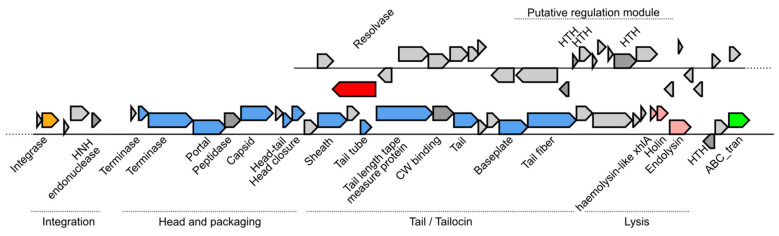
Example of genome organization for a typical “Resolvase” phage/tailocin. The specific prophage represented is from *C. beijerinckii* Group 2 DJ068—AA 004 (Young) genome (IMG Id: 2739367649, contig Ga0171616_11, region shown: 1,137,000–1,181,000). The different gene modules are highlighted above and below the genome map, and genes with known function are highlighted in color (dark red and orange for integration, blue for structural, and light red for lysis genes). The gene highlighted in green in 3’ of the prophage (“ABC_Tran”) represents a typical integration site for these tailocins.

**Table 1 microorganisms-11-02253-t001:** Strains of *C. acetobutylicum, C. beijerinckii* and *C. pasteurianum* held by five of the major international culture collections giving the original number from the Elizabeth McCoy A and B culture collection with the strain origin. The ATCC {862} strain is no longer listed *. The NCIMB 8052 T strain was mis-labeled and is a *C. beijerinckii* strain **.

	*Clostridium acetobutylicum* Species					
McCoy B	Strain Origin	ATCC	NRRL	NCIMB	DSM	JCM
B-3	Weizmann > CSC			6441	1733	
B-5	Weizmann > Speakman			6442		
B-10	Weizmann > Speakman			6443		
B-15	*C. acetonigenum* > Kluyver > Donker > Speakman	862 *	B-528			
B-16	Weyer Type strain	82	B-527	*8052* **	792	19013
B-27	*C. baconyi* > Castell	8529			1738	
B-28	Hall strain	3625	B-529		1737	
B-29	*B. butylicus* > Lister > Thaysen	4259	B-530	619	1731	19012
-	Weizmann > Thaysen			2951	1732	
	***Costridium beijerinckii* Species**					
**McCoy A**	**Strain Origin**	**ATCC**	**NRRL**	**NCIMB**	**DSM**	**JCM**
A8	McCoy isolate > CSC		B-591			19002
A13	McCoy isolate			6444		
A14	McCoy isolate	10132	B-594	8049	1739	19011
A14	McCoy isolate—ATCC 824T contaminant			8052		
A16	McCoy isolate > *C. madisonii*					
A21	*C. butylicum* > Beijerinck > Kluyver		B-593	9380	6423	
A38	*B. butyicus*—FB BB Fernbach > Andrewes		B-596			
A39	*B. fitz* > Pasteur Institute		B-592		6422	19008
A39?	Donker > Reid 39-90		B-466			
A48	*B. bakoni* > Castell	8529				
A51	*C. multifermentans*	3538				
A65	*C. pasterianum* > Winogradski	861				
A67	*C. beijerinckii*—Kluyver	858		11373	1820	1319
A67	*C. beijerinkii* > Kluyver > McCoy > McClung	25752		9362	791	1390
A72	*C. amylobacter* > Haseehoff > Pribram		B-597			
A75	*B. saccharobutyricus* > von Kleeki > Pribram	6015				19003
A77	*C. pasteurianum* > Bezssonoff > Pribram		B-598			
A79	*C. butyricum*—Parazmonski > Pribram	6014				19007
	***Clostridium pasteurianum* Species**					
**McCoy A**	**Strain Origin**	**ATCC**	**NRRL**	**NCIMB**	**DSM**	**JCM**
A5	*C. pasteurianum* > Winogradsy	6013		9486	525	1408

**Table 2 microorganisms-11-02253-t002:** Key genome attributes of the 4 industrial solvent-producing species.

Species	Genome Size (Mb)	% G+C	Genes
*C. acetobutylicum*	4.07–4.18	30.7–30.9	4079–4205
*C. saccharobutylicum*	4.81–5.15	28.4–28.7	4420–4788
*C. saccharoperbutylacetonicum*	6.38–6.69	29.5–29.6	5934–6210
*C. beijerinckii* Group 1	6.01–6.65	29.8–29.9	5056–5668
*C. beijerinckii* Group 2	5.61–6.50	29.5–29.9	4972–5826
*C. beijerinckii* Group 3	5.82–6.48	29.5–30.0	5183–5827
*C. beijerinckii* Group 4	5.97–6.18	29.6–30.0	5326–5491

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
