# Peer review of "Solvent-Producing Clostridia Revisited"

_microorganisms, 2023, doi:10.3390/microorganisms11092253_

Round 1

Reviewer 1 Report

The topic selection is significant, and this work is suitable for the Journal of Microorganisms. However, this manuscript needs a major revision before being published in this journal.

1.        In section 2.6, The butanol fermentation in Taiwan, this description is controversial and would be unacceptable if left uncorrected. “Taiwan” should not be compared with these countries, because “Taiwan” is just one region of China. So the title of section 2.6 “The butanol fermentation in Taiwan” should be changed to “The butanol fermentation in Taiwan(China) or Chinese Taibei”.

2.        Some description is controversial and would be unacceptable if left uncorrected. For example, on page 1 line 45, page 9 line 418, page 14 line 714, and in section 2.6, “Taiwan” should not be compared with these countries, Taiwan should be corrected as Taiwan(China), 

3. This manuscript needs to be reorganized into 3 parts, Section 1, Section 2, and Section 3 can be classified into Part 1:the history of the industrial process. Section 4, Section 5, and Section 6 can be classified into Part 2: The solvent-producing clostridia. Section 7, and Section 8 can be classified into Part 3: Prophages and tailocins. In addition, the reason why the three parts (part 1:the history of the industrial process; part 2: The solvent-producing clostridia; part 3: Prophages and tailorin ) discussed in this manuscript was suggested to be added in the first paragraph of this work.

Minor editing of English language required

Author Response

Reply to reviewers comments attached

Reviewer 2 Report

This review provides a historical/comprehensive analysis of solvent-producing clostridia. The manuscript is well written and is likely to be useful to the scientific community for many years. My only critiques are fairly minor:

1. Any concentrations described as “%” need to have associated descriptor, such as vol/vol, wt/vol, wt/wt, mol/mol, etc.

2. Please consult the journal’s style instruction regarding the presentation of values as numerals vs words. For example, 6 vs six.

3. The butanol toxicity section should include at least a paragraph on efforts that have been used to try and increase organism robustness.

4. It would be useful to include a timeline to accompany the historical descriptions.

5. Line 235 – include the year associated with this dollar value.

6. The section headings “The butanol fermentation in X” headings get repetitive after awhile.

7. Line 953 – “in” does not need to be capitalized

8. In the section about Brazil, there is some variation in font size.

9. Line 1329, the “2” on H2 should be subscript, not superscript.

10. Line 1381 – the author should provide a figure number when referencing the photograph.

11. In Table 1, some values are bracketed or underlined or have an associated asterisk. These designations should be explained.

12. Please check whether organism names in Figures should be italicized.

13. Table 2, units are needed on the “genome” column. Also Table 2, the Group 1 organism name should be italicized.

14. The authors/editors should consider making the tailocin section a stand-alone review.

Author Response

Reply to reviewer's comments attached

Reviewer 3 Report

1.       Part 1: the history of the industrial process is too long, it is recommended to delete some content.

2.       Some wrong writing formats need to be corrected, such as Line 1329, H2,

3.       Page 27, "Lignocellular biomass...". For related content, it is recommended to cite CO2 favors the lipid and biodiesel production of microbial-bacterial granular sludge[J]. Results in Engineering, 2023, 17: 100980.

4.       The content statement in the manuscript is too long and does not highlight the key points. It is recommended to delete some of the content.

5.       Some images that use other literature need to list the relevant literature.

6.       It is suggested that some content irrelevant to the subject of the manuscript be deleted to reduce the word count of this manuscript. For example “Taiwan had also been developed as a major Japanese military base that included around 50 airfields used mainly for pilot training and the navy operated extensively out of the ports around the island.”

7.       Conclusions need to be rewritten: The main conclusions of the study may be presented in a short Conclusions section, which may stand alone.

No

Author Response

Reply to reviewer's comments attached

Round 2

Reviewer 1 Report

these suggestions mentioned before have not been adopted, so it maybe unacceptable.

this paper needs a  revision.

Reviewer 3 Report

1. The author did not modify the reviewer's comments. 2. Some unnecessary political factors are involved. Not consistent with scientific papers. 3. Found plagiarism, academic misconduct.

Need to modify.